# Simulating the atmospheric response to the 11-year solar cycle forcing with the UM-UKCA model: the role of detection method and natural variability

Ewa M. Bednarz[1,a], Amanda C. Maycock[1,2,b], Paul J. Telford[1,2], Peter Braesicke[1,2,c], N. Luke Abraham[1,2] and John A. Pyle[1,2]

[1] Department of Chemistry, University of Cambridge, Cambridge, UK
[2] National Centre for Atmospheric Science - Climate, UK
[a] now at: Lancaster Environment Centre, Lancaster University, Lancaster, UK
[b] now at: School of Earth and Environment, University of Leeds, Leeds, UK
[c] now at: Karlsruhe Institute of Technology, Institute for Meteorology and Climate Research, Karlsruhe, Germany

Correspondence to: Ewa M. Bednarz (e.bednarz@lancaster.ac.uk)

## Abstract

The 11-year solar cycle forcing is recognised as an important atmospheric forcing; however, there remain uncertainties in characterising the effects of solar variability on the atmosphere from observations and models. Here we present the first detailed assessment of the atmospheric response to the 11-year solar cycle in the UM-UKCA chemistry-climate model using a three member ensemble over the recent past (1966-2010). Comparison of the model simulations is made with satellite observations and reanalysis datasets. The UM-UKCA model produces a statistically significant response to the 11-year solar cycle in stratospheric temperatures, ozone and zonal winds. However, there are also differences in magnitude, spatial structure and timing of the signals compared to observational and reanalysis estimates. This could be due to deficiencies in the model performance, and so we include a critical discussion of the model limitations, and/or uncertainties in the current observational estimates of the solar cycle signals. Importantly, in contrast to many previous studies of the solar cycle impacts, we pay particular attention to the role of the chosen analysis method in UM-UKCA by comparing the model composite and a multiple linear regression results. We show that the stratospheric solar responses diagnosed using both techniques largely agree with each other within the associated uncertainties; however, the results show that apparently different signals can be identified by the methods in the troposphere and in the tropical lower stratosphere. Lastly, we examine how internal atmospheric variability affects the detection of the 11-year solar responses in the model by comparing the results diagnosed from the three individual ensemble members (as opposed to those diagnosed from the full ensemble). We show overall agreement between the responses diagnosed for the ensemble members in the tropical and mid-latitude mid-stratosphere-to-lower-mesosphere, but larger apparent differences at NH high latitudes during the dynamically active season. Our UM-UKCA results suggests the need for

long data sets for confident detection of solar cycle impacts in the atmosphere, as well as for more research on possible interdependence of the solar cycle forcing with other atmospheric forcings and processes (e.g. QBO, ENSO).

## 1 Introduction

Incoming solar radiation plays a crucial role in controlling, amongst other things, stratospheric ozone levels and temperature. The quasi 11-year cycle in the solar irradiance has been given particular attention over the last few decades (see the reviews by e.g.: Gray et al., 2010; Haigh, 2010; Solanki et al., 2013, and references therein). Here, we address this topic using the UM-UKCA (Unified Model coupled to the United Kingdom Chemistry and Aerosol model) chemistry-climate model (CCM). Following recent improvements in the model, we present the first detailed analysis of the atmospheric impacts of the 11-year solar cycle simulated in UM-UKCA. In contrast to many previous solar cycle modelling studies in the literature, the novel approach we take is to pay particular attention to the choice of detection method, comparing the model responses diagnosed using both a composite and a multiple linear regression (MLR) method. In addition we investigate how internal atmospheric variability affects the solar responses diagnosed in the model. Our results are particularly relevant to understanding the potential sources of uncertainty in the estimated atmospheric impacts of the 11-year solar cycle forcing both in models (e.g. Mitchell et al., 2015a; Maycock et al., 2018) and observations/reanalysis (Mitchell et al., 2015b; Maycock et al., 2016).

The variation in solar spectral irradiance (SSI) as a function of wavelength is important for determining the atmospheric response to the 11-year solar cycle. Given a typical change in total solar irradiance (TSI) over the 11-year solar cycle of ~1 $Wm^{-2}$, the associated percentage irradiance variability in the visible and infra-red parts of the spectrum is relatively small while the variability in the ultra-violet (UV) region is larger (Fig. S1, Supplement). According to the fifth Coupled Model Intercomparison Project (CMIP5) recommendations (http://solarisheppa.geomar.de/cmip5; Lean, 2000; Wang et al., 2005; Lean, 2009), the SSI at ~220-240 nm varies by ~3-4 % and at ~180 nm by ~10 % (see Fig. S1, Supplement). The SSI variability is even larger for wavelengths below 180 nm, which has important consequences for mesospheric $O_2$ absorption and resulting shortwave heating there (Nissen et al., 2007). It is now well understood that stratospheric UV absorption by ozone increases stratospheric temperatures, while the radiation at wavelengths below ~242 nm is important for ozone production. This solar-induced ozone response provides an additional source of stratospheric heating (Haigh, 1994).

Solar-induced changes in stratospheric ozone over its 11-year cycle of between ~1% and ~5-6% have been reported from the analysis of various satellite (Soukharev and Hood, 2006; Tourpali et al., 2007; Dhomse et al., 2013; 2016; Maycock et al., 2016) and ground-based records (Tourpali et al., 2007). These, alongside changes in incoming solar UV radiation over the 11-year solar cycle, alter stratospheric temperatures (e.g. Gray et al., 2010). In the tropical upper stratosphere, a temperature increase of ~0.7-1.1 K between solar maximum and minimum has been reported from rocketsonde and satellite data (Dunkerton et al., 1998; Ramaswamy et al., 2001; Keckhut et al., 2005; Randel et al., 2009; SPARC, 2010), with somewhat

larger responses found in some reanalysis datasets (Frame and Gray, 2010; Mitchell et al., 2015b). In addition to the temperature and ozone responses in the mid-/upper stratosphere, secondary maxima have been identified in the tropical lower stratosphere, which are often explained to be of dynamical origin (see below).

The most frequently invoked mechanism to explain how the direct solar-induced response in the upper stratosphere can propagate down to the lower atmosphere and affect tropospheric climate is that first proposed by Kuroda and Kodera (2002) and Kodera and Kuroda (2002). In particular, Kuroda and Kodera (2002) reported the development of a positive zonal wind anomaly in the subtropical upper stratosphere for solar maximum during autumn in each hemisphere that is consistent with the strengthened horizontal temperature gradient. In that study, the positive Northern Hemisphere (NH) zonal wind response
developed and propagated poleward and downward during the course of early winter, accompanied by a relative cooling of the polar stratosphere. These early winter anomalies were followed by opposite signed responses in late winter. The authors (see also Kodera and Kuroda, 2002) postulated that the initial strengthening of the vortex near the subtropical stratopause initiates a chain of dynamical feedbacks between atmospheric planetary waves and the zonal mean flow that modulates the strength of the stratospheric jet (thereby influencing an internal mode of high latitude stratospheric variability), with reduced
upward/equatorward wave propagation associated with a stronger and colder polar vortex, and vice versa. This mechanism was developed further by Kodera and Kuroda (2002) who also postulated that the solar-induced modulation of wave activity could impact on the large scale Brewer-Dobson Circulation (BDC), as manifested by the weakening of both the high latitude downwelling and the tropical upwelling in early winter (and the opposite in late winter). The authors noted that this in turn would lead to adiabatic warming and higher ozone levels in the tropical lower stratosphere in early winter, thereby accounting
for the secondary temperature and ozone maxima observed in the region. It is plausible that coupling between the tropical lower stratosphere and tropospheric convection could also play a role in producing solar responses in this region (e.g. Yoo and Son, 2016).

Although there has been much progress in understanding the solar cycle impacts in the atmosphere, there remain uncertainties
in both observational and model derived estimates of the response The observational record covers a relatively short period: only three full 11-year solar cycles have been observed in the satellite (post ~1979, e.g.: SPARC, 2010) period, with a fourth observed cycle currently underway. In addition, there are observational uncertainties associated with the vertical/spatial resolution, changing instruments, as well as scarcity of long-term measurements in certain regions, e.g. the upper stratosphere/lower mesosphere (e.g.: Austin et al., 2008; SPARC, 2010; Mitchell et al., 2015b; Maycock et al., 2016). As a
result, studies have found marked differences between individual datasets in terms of both the magnitude and structure of the stratospheric ozone and temperature responses to the 11-year solar cycle (e.g. Soukharev and Hood, 2006; Mitchell et al., 2015b; Maycock et al., 2016; Dhomse et al., 2016).

Various studies have also assessed the response to an imposed solar cycle forcing in numerical models (e.g.: Haigh, 1999; Matthes et al., 2004; 2013; Austin et al., 2007; Schmidt et al., 2010; Chiodo et al., 2012). While some CCMs simulate solar

responses that broadly resemble the solar cycle signals derived from observations/reanalyses, some studies have found , a marked spread of the solar responses between different models (see e.g.: Austin et al., 2008; SPARC, 2010; Hood et al., 2015; Mitchell et al., 2015a; Maycock et al., 2018). The reasons for that spread are still not properly understood.

Notably, different statistical techniques have been employed to extract the solar-induced responses from observations and model simulations, which can make it difficult to compare results between studies . Some authors composite data into bins representing periods of high/low solar cycle forcing (e.g.: Kuroda and Kodera, 2002; van Loon et al., 2004; Chiodo et al., 2012, and Matthes et al., 2013), while others use more complex statistical regression techniques (e.g. MLR) to account for the effects of different drivers  (e.g:. Frame and Gray, 2010; Misios and Schmidt, 2013; Roy, 2014; Mitchell et al., 2015b; Hood
et al., 2015; Maycock et al., 2016; 2018). Clearly, the use of a simpler composite methodology increases the likelihood for the solar signal to be contaminated with noise and/or the effects of other time-varying drivers, particularly in the case of small sample sizes. On the other hand, the MLR approach usually relies on the individual forcings being independent of one another and combining linearly, which might not be the case in a highly non-linear system like the atmosphere. The choice of analysis method was found to affect the derived surface responses to solar cycle (Roy and Haigh , 2010; 2012;Roy,2014). Another
related factor regarding the uncertainty in atmospheric solar responses is the potential influence of internal atmospheric variability on the signal detection, which will again be more prominent in short records.

In this paper, following recent model improvements, we present the first detailed assessment of the atmospheric response to the 11-year solar cycle simulated by an ensemble of integrations performed with the UM-UKCA chemistry-climate model.
We analyse the simulated atmospheric responses to the 11-year solar cycle in UM-UKCA using both composite and MLR techniques, allowing a direct comparison of the methods In order to understand potential limitations in deriving the solar-induced responses from records comparable in length to the current observations and reanalysis datasets, we analyse the solar cycle responses in the individual ensemble members and compare them. We note that another method for isolating the atmospheric solar cycle response in models is to use idealised time-slice integrations (e.g. Bednarz et al., 2018), but that
approach is less comparable to the behaviour of the real atmosphere and is thus not considered here.

Section 2.1 describes the UM-UKCA model version used and the implementation of the 11-year solar cycle forcing in it. Section 2.2 describes the integrations performed and Sect. 2.3 the observational datasets used as a comparison to the model. Statistical analysis methods are discussed in Sect. 2.4. The yearly mean atmospheric response simulated in the model ensemble
is discussed in Sect. 3, with the winter/springtime NH seasonal response discussed in Sect. 4. Section 5 contrasts the results obtained by using the composites and MLR methodologies. Section 6 focuses on the solar responses found from the individual ensemble members. Finally, Sect. 7 summarises the main results.

## 2 Methodology

### 2.1 The UM-UKCA model

#### 2.1.1 The base chemistry-climate model

We use the UK Chemistry and Aerosol Model coupled to version 7.3 of the Met Office Unified Model (UM-UKCA) in the
HadGEM3-A r.2.0 configuration (Hewitt et al., 2011). The configuration uses prescribed sea surface temperatures (SSTs) and sea-ice. The horizontal resolution is 2.5° latitude by 3.75° longitude, with 60 vertical levels up to ~84 km. The time evolution of dynamical prognostic variables, with the exception of density, and the transport of chemical tracers is carried out within a semi-Lagrangian advection scheme described in Davies et al. (2005). The model includes parameterized orographic and non-orographic gravity wave drag (Scaife et al., 2002; Webster et al., 2003) and simulates an internally generated Quasi-Biannual
Oscillation (QBO; Scaife et al., 2002).

The chemistry scheme used is the extended Chemistry of the Stratosphere scheme (CheS+, e.g. Bednarz et al., 2016). It follows from the Standard Stratospheric Chemistry scheme (CheS) described in Morgenstern et al. (2009). However, unlike the lumping of all halogenated source gases into the three main species ($CFCl_3$, $CF_2Cl_2$, and $CH_3Br$) employed in Morgenstern et
al. (2009), there are 12 halogenated source gases that are considered explicitly. In total, the chemical scheme includes 50 chemical species (out of which 46 are tracers, i.e. are transported by the circulation/mixing) and 195 chemical reactions, including 43 photolysis and 5 heterogeneous reactions. The Fast-JX photolysis scheme used is interactive, i.e. the photolysis rates are calculated accounting for changes in optical depth due to, e.g., ozone, aerosols and clouds (Telford et al., 2013). The scheme covers the wavelengths between 177-850 nm. Above 0.2 hPa (i.e. in the mesosphere), where shorter wavelength
radiation becomes more important, photolysis rates are instead calculated using look up tables (Lary and Pyle, 1991). CheS+ includes only a simple tropospheric chemistry, including emissions of $NO_x$ (in the form of NO), CO and HCHO (Morgenstern et al., 2010). The emissions of ozone depleting substances (ODSs) as well as of $CH_4$, $N_2O$ and $H_2$ are accounted for using lower boundary concentrations. The model-calculated concentrations of $O_3$, $N_2O$, $CH_4$, $CCl_3F$, $CCl_2F_2$, $C_2Cl_3F_3$ and $CHClF_2$ are coupled to the radiation scheme; the specific humidity field from the physical model and prescribed $CO_2$ concentrations
are also used by the radiative scheme.

#### 2.1.2 The representation of the 11-year solar cycle in UM-UKCA

The 11-year solar cycle variability has been implemented in UM-UKCA in both the radiation and photolysis schemes. This is an advance on earlier version of the model that did not include solar cycle variability (see e.g. SPARC, 2010). The method for implementing the solar cycle forcing in the shortwave radiation scheme follows that in the HadGEM1 (Stott et al., 2006) and
HadGEM2-ES models (Jones et al., 2011). The model's radiation scheme (Edwards and Slingo, 1996) has six spectral bands from 200 nm to 10.0 μm (Table 1) in the shortwave spectral region. The yearly mean TSI data used are those recommended for the CMIP5 simulations (Fröhlich and Lean, 1998; Lean, 2000; Wang et al., 2005; Lean, 2009), post-processed to constrain

the mean over 1700-2004 to be 1365 Wm$^{-2}$ (Jones et al., 2011). A fit to spectral data from Lean (1995) is used to account for the change in partitioning of solar radiation into wavelength bins. Table 1 shows how the TSI change of 1.06 Wm$^{-2}$, i.e. the difference between the years 1981 (solar maximum, SMAX) and 1986 (solar minimum, SMIN), is partitioned into the individual bands of the model's shortwave radiation scheme. Using this parametrization, the irradiance in the main UV band (200-320 nm) increases during SMAX by ~0.16 Wm$^{-2}$ (0.56 %). Note that this is ~20 % smaller than the corresponding 200-320 nm spectral irradiance change between these two years (~0.20 Wm$^{-2}$) recommended in the more recent CMIP5 SSI specifications (Lean, 2000; Wang et al., 2005; Lean, 2009).

In the Fast-JX photolysis scheme, the change in partitioning of solar irradiance into wavelength bins is accounted for by scaling the irradiance in the 18 photolysis bins according to the difference in the yearly mean CMIP5 SSI data for the years 1981 (SMAX) and 1986 (SMIN) (see Fig. S1b, Supplement), and the long-term evolution of TSI. As noted above, the Fast-JX scheme is used only for wavelengths between 177-850 nm. At pressures less than 0.2 hPa, i.e. where photolysis rates are calculated using the look up tables, the 11-year solar cycle variability is reflected in the TSI change only, with no modulation of the spectral distribution of solar irradiance.

## 2.2 The UM-UKCA experiments

A three-member ensemble of transient integrations covering 1960-2010 was performed. The first six years of each simulation were treated as spin-up and, therefore, only the 1966-2010 period is analysed below. With the exception of the new implementation of solar cycle variability in the model, the experimental set-up is identical to the UM-UKCA integration shown in the recent SPARC Report on the Lifetimes of Stratospheric Ozone-Depleting Substances, Their Replacements, and Related Species (SPARC, 2013; Chipperfield et al., 2014). The prescribed SSTs and sea-ice follow observations (Rayner et al., 2003). Greenhouse gases (GHGs) follow the SRES A1B scenario (IPCC, 2000) and ODSs follow WMO (2011). The sulphate surface area density (SAD) field recommended for the Chemistry-Climate Model Validation 2 (CCMVal2) models (SPARC, 2006; Eyring et al., 2008; Morgenstern et al., 2010) is prescribed in the stratosphere for the heterogeneous reactions on aerosol surfaces. As with previous versions of UM-UKCA, the coupling of SAD with the photolysis and radiation schemes is not included in this model version.

## 2.3 The observation/reanalysis datasets

Observed estimates of the temperature and zonal wind responses to the solar cycle forcing are derived from the ERA-Interim (ERAI) reanalysis (Dee et al., 2011). We note that the existence of artificial jumps in the upper stratospheric temperature record, which can influence the diagnosed solar cycle response (see e.g. Hood et al., 2015; Mitchell et al., 2015b), was corrected prior to the analysis following the procedure described in McLandress et al. (2014). Although we use only one reanalysis dataset here for comparison with the model, it is important to note that there are quantitative differences in the diagnosed stratospheric responses to solar forcing amongst various current analysis (Mitchell et al., 2015b).

For ozone, we use the SAGE II (Stratospheric Aerosol and Gas Experiment II) ozone data at version 7.0 (Damadeo et al., 2013), which spans the period from October 1984 to August 2005. SAGE II data constitutes the longest continuous record of stratospheric ozone by a single instrument; it is characterised by a good vertical resolution of ~1 km but a sparse horizontal sampling (Soukharev and Hood, 2006; Damadeo et al., 2013; 2014; Hood et al., 2015; Tummon et al., 2015). We use the units of ozone number density, as opposed to volume mixing ratios, in order to avoid uncertainties associated with the choice of temperature record used for the conversion (see e.g. Maycock et al., 2016; Dhomse et al., 2016).

## 2.4 The analysis methods

We use two statistical methods for isolating the atmospheric response to the solar cycle forcing. The first is a simple composite methodology (henceforth referred to as 'composites'). Data from each ensemble member are linearly detrended and three years of data per each SMAX and SMIN are chosen based on the three highest and three lowest yearly mean TSI values in each ~11-year cycle (Fig. 1). This gives 12 years of SMAX and 12 years of SMIN per ensemble member, resulting in 36 SMAX and 36 SMIN years for the full ensemble. A mean SMAX-SMIN difference is calculated and its magnitude scaled to represent a response per 1 Wm$^{-2}$ of the TSI.

The second method is a multiple linear regression (MLR) technique. The MLR code is the same as that used and described in SPARC (2010), with a similar method also used in Bodeker et al. (1998) and Kunze et al. (2016). In this case, the time evolution of a variable, $y(t)$, can be defined as:

$$y(t) = b_{(offset)} \cdot offset + b_{(trend)} \cdot trend(t) + b_{(ESC)} \cdot ESC(t) + b_{(QBO)} \cdot QBO(t) + b_{(QBO\_orth)} \cdot QBO\_orth(t) + b_{(ENSO)} \cdot ENSO(t) + b_{(SAD)} \cdot SAD(t) + b_{(TSI)} \cdot TSI(t) + R(t) \tag{1}$$

The time-varying variables in Eq. 1, apart from the residual, $R(t)$, are input basis functions and the b terms are the corresponding regression coefficients. $b_{(offset)} \cdot offset$ accounts for a mean annual cycle and $b_{(trend)} \cdot trend(t)$ accounts for a linear trend. For the MLR analysis of the UM-UKCA data, $ESC(t)$ is the Effective Stratospheric Chlorine (Eyring et al., 2007) and is defined as the global monthly mean $Cl_y + 60 \times Br_y$ at 20 km, as was done in SPARC (2010) following Newman et al. (2007). $Cl_y$ and $Br_y$ denote total inorganic chlorine and bromine, respectively. The $QBO(t)$ term is defined by the equatorial (1.5°S-1.5°N) zonal mean zonal wind at 50 hPa and the $QBO\_orth(t)$ is a function orthogonal to it. $ENSO(t)$ is calculated from the model's SSTs in the form of the Nino3.4 index (e.g.: Trenberth, 1997), i.e. the zonal and meridional mean SSTs anomaly over the 5°S-5°N 120°W-170°W region, smoothed with a 5-month running mean (here applied 5 times). $SAD(t)$, which represents the volcanic forcing, is defined here as the 30°S-30°N mean of the vertical mean (12-40 km) sulphate SAD field. Note that, as described above, the model configuration includes volcanic aerosols only in the heterogeneous chemistry scheme, and not in radiation or photolysis schemes. The solar cycle forcing is accounted for by the yearly mean TSI value used in the model to

scale the amplitude of the 11-year cycle variability in both the shortwave heating and photolysis schemes (Fig. 1). We note that while some correlation between the trend and ESC functions is plausible, this is considered of secondary importance here as the study aims to isolate the solar response only.

A long-term mean is removed from all basis functions input to the MLR model apart from the offset and SAD(t). The QBO(t), QBO_orth(t) and ENSO(t) functions are additionally detrended. The TSI function is not detrended as the overall trend during 1966-2010 was found to be small and negative (unlike the long-term increase in TSI since the Maunder Minimum, see e.g.: Jones et al., 2011). In order to derive the yearly mean solar response, the MLR analysis is performed using all monthly mean data and the seasonal cycle is accounted for by expanding each of the regression coefficients in a Fourier expansion into two

pairs of sine and cosine functions. For calculating the regression coefficients for the individual months, the MLR analysis is carried out using monthly mean data for each individual month separately with no Fourier expansion. This makes the seasonal calculation more comparable in design to many of the previous MLR studies of the solar response (e.g.: Frame and Gray, 2010; Hood et al., 2013; 2015; Mitchell et al., 2015a; 2015b; Kodera et al., 2016) as well as to the composites calculated here. In addition, when tested on the simulated zonal mean zonal wind and temperature data the approach was found to result in locally

slightly higher $R^2$ values for some of the individual months than $R^2$ for the individual months calculated with the Fourier expansion described above (not shown). An example of a resulting fit and a residual is shown in the Supplement (Fig. S2).

In order to estimate the statistical significance of the derived regression coefficients, the first MLR calculation is followed by a transformation of the regression model (Markus Kunze, pers. comm.; Tiao et al., 1990; Bodeker et al., 1998; SPARC, 2010).

In particular, a second-order autoregressive model for the residuals is used (Eq. 2):

$$R(t) = \rho_1 R(t-1) + \rho_2 R(t-2) + a(t) \qquad\qquad\qquad\qquad (2)$$

where $\rho_1$ and $\rho_2$ are regression parameters and a(t) is a random variable. The fitted values of $\rho_1$ and $\rho_2$ are used to transform the

input basis functions and the input variable y(t), and the MLR is performed again to yield the associated t-test statistics and/or t-test probabilities.

In order to derive solar regression coefficients from the full ensemble, the model output and basis functions from each of the three ensemble members were concatenated into a 135-year-long timeseries and the MLR was performed using the combined

datasets. A similar approach has been adopted in Gray et al. (2013) and Hood et al. (2013).

For consistency with the model, the MLR analysis of ERAI and SAGE II data also employs TSI as the regressor for the solar cycle forcing. However, unlike the yearly mean TSI timeseries that forces the model, the timeseries chosen here is that originally recommended for the CMIP5 models (http://solarisheppa.geomar.de/cmip5; Fröhlich and Lean, 1998; Lean, 2000;

Wang et al., 2005; Lean, 2009; Fig. 1), varying on a monthly basis We note that a number of proxies for solar forcing has been used in the literature (see Gray et al., 2010, for details), with one of the most common being the 10.7 cm solar radio flux (F10.7). Figure 1 (blue and red curves) illustrates that there is a good degree of correlation between the TSI and F10.7 proxies, in particular on longer timescales (R=0.81, with 1 $Wm^{-2}$ = 101 (±8) solar flux units, SFU).

For simplicity, the observed solar responses in ERAI and SAGE II discussed here are derived from MLR only. In this case, the volcanic regressor used, SAD(t), is the same as in the model. The QBO(t), QBO_orth(t) and ENSO(t) were calculated in the same way as in the model but using the ERAI zonal wind and SST data (Dee et al., 2011). ESC(t) is replaced with EESC(t) (Equivalent Effective Stratospheric Chlorine), which estimates stratospheric $Cl_y$ and $Br_y$ from their tropospheric source gases, here assuming the atmospheric circulation with the age of air spectrum with the mean of 3 years (as was done in SPARC, 2010, following Newman et al., 2007). The MLR analysis is performed over 1979-2008 for ERAI, and over October 1984 to August 2005 for SAGE II.

## 3 The ensemble yearly mean response in UM-UKCA

This section focuses on the yearly mean atmospheric response to the 11-year solar cycle forcing simulated in the ensemble of the transient UM-UKCA integrations. In particular, the yearly mean changes in shortwave heating rates, temperature, ozone and zonal wind from the combined ensemble are discussed; the model responses are derived using both MLR and composites and, where available, compared with the reanalysis or observations.

### 3.1 Shortwave heating rates

Figure 2a shows the yearly mean tropical mean (25ºS-25ºN) shortwave heating rates response to the 11-year solar cycle forcing in the UM-UKCA model, expressed as a response per $Wm^{-2}$ change in TSI. The response maximises in the tropical stratopause region, reaching up to ~0.16 K $day^{-1}$/$Wm^{-2}$ and 0.14 K $day^{-1}$/$Wm^{-2}$ for the MLR and composite methods, respectively. Notably, this solar-induced modulation constitutes a relatively small fraction of the absolute short wave heating rates in this region (~1.5 % near 50 km). The magnitude of the MLR-derived maximum is somewhat larger than in the composites, although the two are not significantly distinguishable, in a statistical sense, given the estimated uncertainties in each.

### 3.2 Temperature and zonal winds

### 3.2.1 The tropical upper stratospheric temperature response

The corresponding yearly mean tropical temperature response to the solar cycle forcing is shown in Fig. 2b, compared with the ERAI reanalysis. The MLR tropical temperature response in UM-UKCA maximises above the stratopause at ~0.8 K/$Wm^{-2}$. In comparison, the tropical mean ERAI response maximises in the upper stratosphere at ~1-1.1 K/$Wm^{-2}$ (in agreement with

previous ERAI studies, e.g.: Mitchell et al., 2015b; Hood et al., 2015; Kodera et al., 2016). Compared to ERAI the UM-UKCA simulated temperature maximum thus occurs at higher altitudes and is ~25% smaller.

A number of factors could explain the apparent underestimation of the maximum tropical temperature response in UM-UKCA as compared with ERAI. One contributing factor may be the broadband shortwave heating scheme, with only one band in the UV and two in the visible parts of the spectrum (Section 2.1.2). Nissen et al. (2007) showed that decreasing the number of spectral bands in the UV/visible range from 49 to 6 can result in an underestimation of the stratopause shortwave heating response to the 11-year cycle by ~20 %. In addition, only the absorption of solar radiation by ozone is considered in the first (UV) shortwave spectral band (Table 1), thereby neglecting the absorption by molecular oxygen. Also, limiting the shortwave heating scheme to the wavelengths higher than 200 nm excludes changes in the mesospheric absorption by oxygen near the Lyman-α line (121.6 nm), where percentage irradiance changes during the solar cycle can be particularly large (Lean, 2000; Nissen et al., 2007).

Another factor that is likely to be important for the magnitude of the upper stratospheric temperature response is the fact that the model has used a relatively modest modulation of SSI. There has been considerable uncertainty associated with the solar cycle modulation of SSI due to the shortage of long-term satellite measurements, with marked differences between the individual available datasets (e.g. Harder et al., 2009; Dhomse et al., 2013; Ermolli et al., 2013). We also note that whilst being consistent with the design of the HadGEM2-ES model (Jones, et al., 2011), the current implementation of the solar cycle forcing in the model's radiation scheme results in an underestimation of the UV changes in the 200-320 nm band by around ~20 % compared to the CMIP5 SSI recommendations (Section 2.1.2).

Lastly, there are large uncertainties in the reanalysis datasets in the upper stratosphere due to the scarcity of long-term measurements (McLandress et al., 2014; Long et al., 2017). As a result, large differences exist between reanalyses in both the structure and magnitude of the upper stratospheric/lower mesospheric temperature response to the solar cycle (Mitchell et al., 2015b). A somewhat smaller temperature response was estimated from the stratospheric sounding unit satellites (SPARC, 2010; Randel et al., 2009[1]), although this could be related to their relatively poor vertical resolution (Gray et al., 2009).

We note that even models forced with identical SSI variations still show relatively broad ranges of solar cycle temperature responses (e.g. Austin et al., 2008; SPARC CCMVal, 2010; Mitchell et al., 2015a). SPARC CCMVal (2010) showed that model performance in simulating the direct radiative response to a change in solar irradiance alone could not fully account for the spread of simulated temperature responses, suggesting some contribution of indirect dynamical processes. This aspect will

---

[1] Up to ~0.6-0.7 K/100 SFU, SPARC, 2010; up to ~1 K/125 SFU, Randel et al., 2009. Recall that 1 Wm$^{-2}$ ≈ 100 SFU, Section 2.4.

be investigated further in Sect. 6, where the solar response found from the individual ensemble members is considered. Lastly, the differences in stratospheric temperature responses could be related to solar ozone responses, which is discussed in Section 3.3.

### 3.2.2 The tropical lower stratospheric temperature response

The annual mean stratospheric temperature response to the 11-year solar cycle forcing in UM-UKCA diagnosed using MLR decreases in magnitude with decreasing altitude and does not show a secondary temperature maximum as found in ERAI in the lower stratosphere (Fig. 2b). According to the mechanism postulated by Kodera and Kuroda (2002), the anomaly results from the solar-induced modulation of the BDC brought about by the strengthened horizontal temperature gradient during the dynamically-active season in each hemisphere. Thus, differences between the model and reanalysis could result from the

differences in the dynamical responses. Accordingly, while ERAI shows yearly mean strengthening of the extratropical zonal winds in both hemispheres (Fig. 4c), no such yearly mean westerly anomalies were found in UM-UKCA (Fig. 4b). Instead, the yearly mean UM-UKCA simulated response is dominated by the regions of a relative zonal wind deceleration. Therefore, any strengthening of the stratospheric vortex and the associated reduction in the large scale circulation in UM-UKCA is too weak and/or short lived to have an impact on the tropical upwelling that would be visible in the yearly mean. Possibly, the

coupling of the ocean and tropical convection, which may also play a role in influencing the tropical lower stratosphere (e.g. Yoo and Son, 2016), is also not adequately represented in the model set-up with prescribed (although observationally derived) SSTs. In the case of the Northern Hemisphere, a more detailed seasonal analysis of the simulated dynamical response is presented in Section 4.

One challenge for attributing signals in the tropical lower stratosphere is possible aliasing between the effects of solar forcing with other natural forcings and processes, e.g. volcanic forcing, QBO or ENSO (e.g.: Lee and Smith, 2003; Marsh and Garcia, 2007; Smith and Matthes, 2008; Chiodo et al., 2014; Mitchell et al., 2015b). Chiodo et al. (2014) used the WACCM model and found that the secondary tropical temperature maximum widely attributed to the 11-year solar cycle over the reanalysis period largely disappeared if volcanic eruptions were not included in the model. In this version of UM-UKCA the stratospheric

aerosols are coupled only to the heterogeneous chemistry scheme, and not to the photolysis or radiative heating schemes; the relative overlap between the elevated values of the aerosol SAD index following the main volcanic eruptions and the years selected as composite SMAX years is also relatively small here (not shown). Therefore if aliasing with volcanic eruptions was an important contributor to the lower stratospheric temperature maximum in reanalysis, this effect would not be reproduced in UM-UKCA.

### 3.2.3 The mid-latitude troposphere

In the troposphere, ERAI shows a small warming (~0.1-0.2 K/Wm$^{-2}$) in the mid-latitudes on both hemispheres (Fig. 3c), accompanied by a weakening of the extratropical zonal winds at ~30º in the troposphere and lower stratosphere, alongside somewhat weaker opposite sign responses in the mid-latitudes (Fig. 4c). These tropospheric dipole patterns represent a weakening and poleward shift of the mid-latitude jets and an associated expansion of the Hadley circulation (Haigh et al., 2005; Haigh and Blackburn, 2006).

In agreement with ERAI, UM-UKCA simulates a statistically significant warming of up to ~0.2 K/Wm$^{-2}$ in the NH mid-latitude troposphere, detectable for both analysis methods (Fig. 3a-b). This is accompanied by a weakening of the NH extratropical zonal wind (MLR), which is broadly similar to ERAI albeit smaller in magnitude and without a strong accompanying westerly response in the mid-latitudes (Fig. 4a-b). Interestingly, the modulation of the NH subtropical jet in UM-UKCA occurs in the absence of a strong yearly mean tropical warming in the lower stratosphere, which has been shown to be a driver of the tropospheric wind responses (Haigh et al., 2005; Simpson et al., 2009), and/or strong yearly mean stratospheric westerly anomalies. However, Misios and Schmidt (2013) showed that a weakening and poleward shift of the subtropical jets that projects onto the timescale of solar cycle forcing could be reproduced in a model forced only with the prescribed observationally-derived SSTs/sea-ice. It is therefore plausible that the solar signal, or other variability, found in the prescribed observed SSTs could contribute to the diagnosed tropospheric wind changes in UM-UKCA. We note, however, that unlike the hemispherically symmetric tropospheric zonal wind and temperature response in ERAI, UM-UKCA does not capture such poleward shift of the yearly mean tropospheric jet or a mid-latitude warming in the SH. The role of prescribed SSTs in the diagnosed solar cycle response is further discussed in Sect. 6 in the context of the solar responses found in the individual ensemble members.

### 3.3 Ozone

### 3.3.1 Total ozone column

The UM-UKCA MLR results show a yearly mean global mean total column ozone response of around ~6 DU/Wm$^{-2}$ (Table 2). In comparison, total ozone column responses over the solar cycle of a few DU in the tropics/mid-latitudes have also been reported from observations (Randel and Wu, 2007; SPARC, 2010; Lean, 2014), but with some differences between the individual datasets or their different versions (Randel and Wu, 2007; Lean, 2014).

### 3.3.2 The tropical/mid-latitude stratosphere

In the tropical mid-/upper stratosphere, the UM-UKCA model simulates an ozone response of up to ~2.0-2.5 %/Wm$^{-2}$ (Fig. 5a-b). The maximum ozone response in UM-UKCA is smaller than that diagnosed in SAGE II (up to 3-3.5 %/Wm$^{-2}$, Fig. 5c; see also Soukharev and Hood, 2006; Randel and Wu, 2007; Gray et al., 2009; Dhomse et al., 2013; 2016; Maycock et al.,

2016), and is more uniform over the mid-stratosphere compared to the more peaked structure in the upper stratosphere in SAGE II. It also maximises at lower levels than in the observations (Fig. 2c; Fig. 5). The relatively lower altitude of the tropical ozone maximum compared to satellite observations is a common feature amongst various CCMs (Maycock et al., 2018). We note that differences exist between the magnitudes as well as the structures of the ozone responses found in different satellite products and/or their different versions (Soukharev and Hood, 2006; Dhomse et al., 2013; 2016; Maycock et al., 2016).

The lower altitude of the maximum ozone response in UM-UKCA compared to observations could contribute to the lower magnitude of the tropical temperature response. As noted in Section 3.2.1., there are large differences between the different SSI datasets (e.g. Ermolli et al., 2013) and the UM-UKCA model is forced with relatively modest modulation of SSI at the lower end of the current estimates. Yet, the recent sensitivity study in Maycock et al. (2018) suggested a CCM forced with larger SSI variability would also produce a maximum tropical ozone response to solar forcing at a relatively lower altitude than in satellite observations. The relatively lower altitude of the UM-UKCA model response could also arise due to model deficiencies, e.g. in the photolysis code, and/or uncertainties in the observed response. Regarding the performance of the model photolysis code, Sukhodolov et al. (2016) found a small (~0.2-0.3 %) underestimation of the ozone response to the solar cycle in the upper stratosphere in the Fast-JX code used in UM-UKCA compared with line-by-line calculations[2]; this gave rise to only a small temperature underestimation of ~0.05 K. Thus, it appears that the details of the model photolysis scheme alone cannot explain the apparent differences in the simulated tropical upper stratospheric ozone response compared to SAGE II measurements. With respect to observational uncertainties, it is important to understand that our chemistry-climate model simulates, by definition, an ozone response to the solar cycle forcing that is consistent with the imposed SSI variation and the associated changes in temperature and transport. The modelled impact on stratospheric temperatures of the increased ozone levels under higher solar cycle activity will thus inevitably differ from that simulated by general circulation models that use prescribed observationally-derived ozone changes (Maycock et al., 2018). The role of the representation of the solar-ozone feedback is discussed in detail using specially designed sensitivity experiments in Bednarz et al. (in prep.).

## 4 The ensemble NH seasonal response in UM-UKCA

### 4.1 Temperature and zonal wind

The previous section analysed the yearly mean zonal wind and temperature responses to the 11-year solar cycle simulated in UM-UKCA. The following section presents an analysis of the seasonal evolution of the diagnosed responses with a focus on the NH winter season. Figure 6 shows the November-April monthly mean NH temperature responses in both UM-UKCA and ERAI, and Fig. 7 the associated monthly mean changes in zonal wind.

---

[2] For the solar cycle variability given by COSI SSI, represented as deviations in the photolysis rates of $O_2$, $O_3$ and $H_2O$ (see Sukhodolov et al., 2016, for details).

### 4.1.1 The ERAI reanalysis

The enhanced solar radiation in November results in warming of the tropical/subtropical upper stratosphere (here up to ~1-1.4 K/Wm$^{-2}$) and a region of cooling in the polar latitudes (here up to ~7 K/Wm$^{-2}$). The enhanced horizontal temperature gradient is commensurate with a positive zonal wind anomaly in the subtropics (as well as near the mid-/high latitude stratopause), persisting in the subtropics throughout autumn and winter. In mid-winter (January) it appears to develop and extend into the mid-/lower stratosphere and troposphere, in agreement with the postulated dynamical mechanism involving wave-mean flow interactions (Kuroda and Kodera, 2002; Kodera and Kuroda, 2002). Notably, the zonal wind response in ERAI in January is only statistically significant near the subtropical stratopause as well as in a narrow band in the troposphere at ~60°N. In contrast, none of the changes in the stratospheric polar vortex strength or the high latitude temperature are found to be highly statistically significant, similar to the three-reanalysis-mean results of Mitchell et al. (2015b). This reflects the presence of large interannual variability in the NH winter stratosphere resulting in large uncertainties in extracting an observed solar response over a relatively short time period (see also Sect. 6). In the tropical lower/mid- stratosphere (100-10 hPa), there is a region of warming between November and January, which contributes to the yearly mean secondary temperature maximum discussed in Sect. 3.2.

Later in February, the strengthening/warming of the stratospheric vortex in ERAI is replaced by a stronger and statistically significant response of opposite sign. The February sign reversal has been found to be a robust feature of the various reanalysis data sets available (e.g.: Matthes et al., 2004; Frame and Gray, 2010; Hood et al., 2015; Mitchell et al., 2015b; Kodera et al., 2016). Despite the sign reversal in the mid-/high latitudes, the westerly zonal wind anomaly near the subtropical stratopause persists throughout the winter into spring. It then amplifies and descends down to the stratosphere and troposphere in April, accompanied by an anomalous high latitude cooling (where it persists at reduced magnitude until ~June, not shown, see also Mitchell et al., 2015b). As pointed out by Mitchell et al. (2015b), despite the robustness of the response, these springtime anomalies have largely been ignored in the literature. In general, the ERAI results shown here agree with the previous reanalysis studies (e.g.: Matthes et al., 2004; Frame and Gray, 2010; Mitchell et al., 2015b; Hood et al., 2015; Kodera et al., 2016).

### 4.1.2 The UM-UKCA model

The UM-UKCA results in November show a qualitatively similar but weaker pattern of temperature changes to ERAI consisting of a relative warming in the tropical upper stratosphere/lower mesosphere (up to ~ 1 K/Wm$^{-2}$) and a cooling in the polar region (of up to ~1.5-2.5 K/Wm$^{-2}$). The simulated strengthening of the stratospheric vortex extends down to the troposphere. The tropospheric response is stronger and more significant in the composites, and shows ~1 ms$^{-1}$/Wm$^{-2}$ strengthening of the zonal wind at ~60°N accompanied by a weakening in the subtropics. A suggestion of a small warming

can be seen in the tropical lower stratosphere. However, this is weaker and forms a part of a general warming throughout the depth of the tropical stratosphere than the larger and more localised response in ERAI (November-January).

In December, both the temperature and zonal wind responses simulated in UM-UKCA weaken compared to November. While some cooling remains in the polar lower stratosphere, an opposite sign response develops in the upper stratosphere aloft. In mid-winter (January-February), the polar warming magnifies and descends, alongside the simulated weakening of the stratospheric jet. While the easterly anomaly propagates down through the lower mesosphere and stratosphere, the westerly response redevelops near the subtropical stratopause. It then appears to propagate poleward and, to some extent, down in the stratosphere in spring, reversing sign in April. However, the anomalies derived from MLR and composite analysis are not highly statistically significant in April, suggesting a small signal-to-noise level.

### 4.1.3 Discussion

The evolution of the solar cycle response in the NH high latitudes in winter in UM-UKCA shows a shift in the timing of the UM-UKCA modelled responses towards earlier in the winter. In particular, the strengthening of the stratospheric vortex and its extension to the troposphere were simulated in UM-UKCA in November, with the sign reversal occurring most prominently in January. In ERAI, the indication of the poleward propagation of the subtropical westerly response can be seen later (January) with the pronounced sign reversal occurring rapidly in February. In February, the modelled anomalies agree well with the diagnosed ERAI anomalies. In addition to the different timing of the zonal wind response to ERAI, the simulated westerly response diagnosed from monthly mean data appears first at higher latitudes than in ERAI, thereby not clearly reproducing the poleward/downward propagation seen in the reanalysis (although the model does simulates a significant dynamical response to the imposed solar forcing at high latitudes comparable in magnitude to that diagnosed in ERAI). We also note that the westerly anomalies near the subtropical/mid-latitude stratopause found in ERAI from late autumn till spring are generally much stronger and longer-lived in the reanalysis than in UM-UKCA. Lastly, the UM-UKCA model integrations do not reproduce the westerly zonal wind anomalies found in the reanalysis in the mid-/late springtime (April onwards) stratosphere. Regarding the temperature anomalies simulated in the tropical lower stratosphere, while there is some general warming throughout the tropical stratosphere in UM-UKCA in November (i.e. when the largest westerly anomaly occurs in the mid-/high latitude stratosphere), the temperature increase in the tropical lower stratosphere is very small and does not form a distinct maximum as in ERAI.

The differences in the timing of the UM-UKCA modelled zonal wind and temperature responses compared with the reanalysis (Section 4.1) could be related to a positive bias in the background zonal wind climatology in UM-UKCA (not shown). As pointed out by Kodera and Kuroda (2002), the mechanism behind the high latitude response to the solar cycle forcing relies on the non-linear interactions between the planetary waves and the mean flow and, therefore, is influenced by the climatological mean state. It is well known that the propagation of planetary waves through the mid-/high latitude stratosphere

is strongly dependent on the strength and direction of the background zonal wind, with the planetary waves propagating only if the winds are westerly and not too strong (Charney and Drazin, 1961; Andrews et al., 1987). The importance of a realistic climatology in models for reproducing the observed dynamical response to the solar cycle forcing has also been suggested by other studies, e.g. Kodera et al. (2003); Matthes et al. (2004); Rind et al. (2008); Schmidt et al. (2010); Chiodo et al. (2012). The differences in timing could also be related to errors in the prescribed SSI forcing in UM-UKCA, which may not be representative of the true SSI variability (Ermolli et al., 2013).

## 4.2 Mechanism

As discussed in the introduction, Kuroda and Kodera (2002) and Kodera and Kuroda (2002) postulated that the initial solar-induced enhancement of the horizontal temperature gradient in autumn and the associated strengthening of the subtropical upper stratospheric jet initiate a chain of feedbacks between planetary waves and the mean flow that modulates the evolution of the polar vortex, as well as the BDC, throughout the dynamically active season. A mechanism consistent with that was later shown to operate in a number of modelling studies (e.g. Matthes et al., 2004; 2006; Chiodo et al. 2012).

The changes in zonal mean circulation simulated in UM-UKCA discussed above are associated with consistent changes in wave propagation/breaking. In particular, the development of the positive zonal wind response to higher solar cycle forcing in autumn (November, Fig. 7) is associated with a decreased propagation of planetary waves through the stratosphere (not shown) and a reduction in wave breaking throughout the mid-/high latitude stratosphere and lower mesosphere (divergence of EP flux of up to ~2.5 ms$^{-1}$day$^{-1}$/Wm$^{-2}$, Fig. 8). This reduction in the deposition of eddy heat and momentum accelerates the zonal wind, which should feed back on planetary waves and result in even less wave propagation and breaking. So, there is a two-way interaction between the waves and zonal wind, as also discussed in Kuroda and Kodera (2002), Kodera and Kuroda (2002), Matthes et al. (2004; also 2006) and Chiodo et al. (2012).

Later in December and January, UM-UKCA simulates an increase in the mid-/high latitude wave propagation (not shown) and increased wave breaking under higher solar cycle forcing in December and January, which is particularly evident in the upper stratosphere/lower mesosphere (see the negative divergence of EP flux, i.e. convergence, of up to ~3.5-4 ms$^{-1}$day$^{-1}$/Wm$^{-2}$, Fig. 8). As this decelerates the zonal wind, the results are consistent with the considerable weakening of the westerly zonal wind response in December, compared to November, and its sign reversal in January. In the subtropics, the emergence of a positive zonal wind anomaly near the stratopause in January is associated with the region of a relative EP flux divergence that develops over the mid-/high latitudes in February, consistent with the strengthening of the subtropical westerly signal in that month and its apparent poleward/downward propagation in March. However, the amplitudes of the EP flux divergence anomalies in late winter and early spring are generally smaller than earlier and not highly statistically significant.

Changes in planetary wave propagation and breaking in the high latitudes modulate the BDC. As expected, in November the strengthening of the stratospheric jet and the accompanying reduction in the mid-/high latitude wave breaking under higher solar cycle forcing reduce downwelling in the lower/mid-stratosphere over the polar cap (Fig. 9); this in turn contributes to the high latitude cooling modelled in the region (Fig. 6). In the tropics, we find a suggestion of a small reduction in upwelling in the lower stratosphere (below ~10 hPa) in UM-UKCA, as postulated by the mechanism of Kodera and Kuroda (2002); however, the $\overline{w}^*$ response, particularly at the altitudes of ~100-30 hPa, is very small and not highly statistically significant, indicating that the postulated mechanism is not robustly reproduced in the model. A more statistically robust response was found in the mid-latitudes, possibly indicating a relative shift in the downwelling region towards the extratropics. In December and January, the sign of the model responses reverses

Thus, the UM-UKCA simulated changes in planetary wave breaking and $\overline{w}^*$ in the high latitude stratosphere agree with the postulated mechanism (Kuroda and Kodera, 2002; Kodera and Kuroda, 2002), albeit with some differences in the timing of the responses, as well as with the modelling results of Matthes et al. (2004) and Chiodo et al. (2012). Regarding the tropical lower stratosphere, while some indications of solar-induced upwelling and temperature anomalies are found that are consistent in terms of a sign with the BDC changes in the Arctic region, the magnitudes of the UM-UKCA tropical upwelling (and temperature) anomalies are very small in the lower stratosphere and not strongly statistically significant, indicating that the mechanism is not robustly reproduced in the model.

## 5 The role of detection method: composites vs MLR

Here we compare the UM-UKCA responses diagnosed by the composite and MLR methods. Model and reanalysis based studies in the published literature have adopted both of these methods but they are rarely applied together. The application of both methods to the same model simulations enables a clean comparison of the diagnosed responses.

### 5.1 Ensemble yearly mean temperature and zonal wind response

A comparison between the UM-UKCA tropical temperature responses derived using composites and MLR shows a good agreement between the magnitudes of the detected upper stratospheric/lower mesospheric temperature responses (Fig. 3). However, unlike the weak positive MLR temperature response in the tropical lower stratosphere, composites yield a small statistically insignificant cooling of up to 0.2-0.3 K/Wm$^{-2}$ in that region. This is likely to be a manifestation of contributions from other forcings that affect lower stratospheric temperatures (e.g. QBO, ENSO) and interannual variability. Interestingly, in the troposphere, while no significant temperature response was detected with MLR in the tropics, composites yield a small, albeit locally statistically significant,warming of up to ~0.2 K/Wm$^{-2}$. A large part of this temperature dipole structure seen in composites in the tropical troposphere/lower stratosphere is attributed to the residual term in our MLR analysis, with smaller contributions from the influence of ENSO and QBO (see Supplement S1 and Fig. S3).

Similarly, while the composite and MLR zonal wind responses agree, within the uncertainty estimates, in the extratropical stratosphere, some apparent differences were found in the tropical stratosphere and in the troposphere (Fig. 4). Near the equator, there is a strong contribution from the QBO to the composite response, which is better separated from the solar component in the MLR analysis (Supplement S1 and Fig. S4). In the troposphere, the UM-UKCA composites suggest an equatorward shift of the SH subtropical jet and a weakening of the polar night jet extending down to the surface at ~60°S, in qualitative agreement with ERAI (MLR). In contrast, no significant tropospheric response in the SH was found in UM-UKCA using MLR analysis. As with the tropospheric temperature response, the apparent discrepancy between the MLR and composite responses in the troposphere is related to the fact that parts of the latter are attributed to the residual and, to a smaller extent, ENSO terms in the MLR model (Supplement S1 and Fig. S4).

Assuming that the individual forcings are indeed completely independent from each other and linear, the results suggest that MLR achieves a better level of separation of contributions from other processes and interannual variability not directly related to the solar forcing, therefore minimising the effects of aliasing/noise. However, the impacts of individual forcings may not necessarily be independent from one another and additive. A number of studies suggested that coupling of the solar cycle forcing with, e.g., QBO and/or ENSO, could be important (e.g. Salby and Callaghan, 2000; Pascoe et al., 2005; Labitzke et al., 2006; Haigh and Roscoe, 2006; 2009; Roscoe and Haigh, 2007; Camp and Tung, 2007; Kuroda, 2007; Lu et al., 2009; Calvo and Marsh, 2011; Roy and Haigh, 2011; Matthes et al., 2013). Therefore, we refrain here from judging unambiguously which technique performs better in the detection of the solar responses. More importantly, we stress that the differences between the composite and MLR responses found in the troposphere and in the tropical lower stratosphere, although not statistically significant, illustrate that the use of only one of these techniques could lead to somewhat different conclusions with regard to the solar cycle response and, therefore, highlight that care needs to be taken when analysing solar responses in this region.

## 5.2 Ensemble yearly mean ozone response

Regarding the yearly mean ozone responses in the full ensemble, we find that the total column ozone responses derived in various regions are somewhat higher for MLR than for composites, but agree to within the estimated uncertainty ranges (Table 2). Similarly, the spatial patterns of the MLR and composite ozone responses generally agree with each other in large part of the stratosphere. The main differences (albeit still not statistically significant) occur in the tropical stratosphere. In the lower stratosphere, in contrast to the weak positive MLR response, the composites suggest an ozone decrease of up to ~3-3.5 %/Wm$^{-2}$, consistent with the small temperature decrease found in that region (Fig. 4). As with the temperature and zonal winds, aliasing with other natural forcings could contribute to the tropical lower stratospheric ozone changes in the composites, as suggested by the studies of, e.g., Lee and Smith (2003); Marsh and Garcia (2007); Smith and Matthes (2008) and Mitchell et al. (2015b).

## 5.3 Ensemble NH seasonal response

In general, the monthly mean NH high latitude temperature and zonal wind responses in UM-UKCA derived using composites and MLR during winter/spring are found to be qualitatively and quantitatively similar within the associated uncertainties (Figs. 6 and 7), with somewhat stronger and more significant strengthening of the tropospheric zonal wind at ~60°N found from composites in November.

## 6 Solar cycle response over the recent past in the individual ensemble members

Uncertainties exist regarding the atmospheric response to the 11-year solar cycle forcing. So far, we have focused on the solar cycle response in UM-UKCA across all three ensemble members for the recent past combined. However, observational/reanalysis records such as ERAI or SAGE II represent only a single realisation of the real atmosphere. In order to gain knowledge about underlying variability and, therefore, understand potential issues in deriving solar responses from records comparable in length to the current observations and reanalysis dataset, it is informative to examine the solar responses simulated in each individual ensemble member separately.

The annual mean temperature and ozone responses derived using MLR for the individual ensemble members (denoted ENS1, ENS2 and ENS3) are shown in Fig. 10 and 11, respectively. A reasonable, albeit not perfect, degree of agreement exists between the individual members regarding the tropical mean anomalies: the estimated lower mesospheric temperature maxima range between ~0.7-0.85 K/Wm$^{-2}$ (Fig. 10d) and the mid-stratospheric ozone maxima between ~2-3 %/Wm$^{-2}$ (Fig. 11d). In both cases, this intra-ensemble spread lies within the associated statistical confidence intervals. Yet, the small differences in the vertical profiles illustrate that natural variability cannot be entirely neglected. In the tropical lower stratosphere, while all three ozone responses are characterised by a broad uncertainty range and individually are not statistically significant, we find both a member which shows no suggestion of a secondary ozone maximum in the region (ENS1), as well as another one

(ENS2) that shows an ozone increase of ~3 %/Wm$^{-2}$ that broadly resembles a secondary tropical ozone maximum similar to that found in observational datasets.

The apparent discrepancy between the ensemble members in the diagnosed annual mean responses is largest in the high latitudes, in case of the temperatures particularly in the NH. This mainly reflects the spread of the anomalies derived during the dynamically active season (see below). Interestingly, one member (ENS3) shows an annual mean stratospheric temperature response whose main features resemble the ERAI reanalysis (Fig. 3c). These include the stratopause maximum clearly peaking near the equator (albeit higher up than in ERAI), a strong warming in excess of 1 K over the NH polar mid-stratosphere, as well as a suggestion of another temperature maximum in the SH polar stratosphere below 10 hPa. Unlike in ERAI, however, a strong secondary temperature maximum in the tropical lower stratosphere is not reproduced in ENS3.

As discussed in Sect. 4, a strengthening of the NH polar vortex (up to ~7 ms$^{-1}$/Wm$^{-2}$) in autumn (November) was diagnosed from the MLR analysis across the full ensemble (Fig. 7). When the integrations are analysed separately, the underlying variability is high from November throughout the rest of the dynamically active season and, consequently, many of the anomalies found in the single ensemble members are not found to be highly statistically significant (Fig. 12 as well as Fig. S5, Supplement). In November (Fig. 12), only two out of the three ensemble members show the westerly zonal wind anomaly that was found when analysing the full ensemble. Moreover, even for these two members, the magnitudes of the derived responses differ considerably (although not statistically significantly), with the ensemble member ENS2 showing a response of up to ~16 ms$^{-1}$/Wm$^{-2}$ and ENS3 a smaller ~8 ms$^{-1}$/Wm$^{-2}$. In mid-winter (January and February), all members suggest strengthening of the zonal wind in the extratropical upper stratosphere/lower mesosphere and a weakening in the high latitudes, in qualitative agreement with the response in the full ensemble. Again, the magnitudes of these anomalies differ between the members, although they agree within the estimated uncertainty ranges. In comparison, ERAI show a strong and fairly robust reversal of the high latitude zonal wind response in February, with a strong easterly anomaly commonly found in that region (Fig. 7, as well as Matthes et al., 2004; Frame and Gray, 2010; Hood et al., 2015; Mitchell et al., 2015b; Kodera et al., 2016). As illustrated in Fig. 12, while one of our ensemble members (ENS2) shows a statistically significant easterly response of ~18 ms$^{-1}$/Wm$^{-2}$ in that month, the analogous easterly anomalies found in the remaining two members are comparatively small and not statistically significant. The intra-ensemble agreement somewhat improves in late winter and early spring, with all members indicating a poleward and downward propagation of the westerly response in March, followed by an easterly response in April (Fig. S5, Supplement).

Possibly, the solar cycle signal in the UM-UKCA model is smaller than in the real world, owing to e.g. the relatively weak SSI forcing (Sections 3 and 4). The model integrations also show higher interannual variability (i.e. standard deviation) of the NH zonal wind in parts of the stratosphere, e.g. the upper stratosphere around 60˚N in autumn/winter, than ERAI (not shown). Both of these effects will impact on the signal-to-noise ratio and the detectability of the solar cycle signal (Scaife and Smith,

2018). Nevertheless, apparent discrepancies between solar responses derived from individual ensemble members have also been noted in other studies (e.g. Austin et al., 2008; Chiodo et al., 2012). Interestingly, Hood et al. (2013) analysed the sea level pressure responses to the solar cycle forcing simulated in the north Pacific and found that limiting the length of their otherwise 16 solar-cycle-long simulations to ~9.5-cycle-long sub-periods can give responses that are apparently stronger and

more significant in some of the sub-periods than in the full simulation (or in the other sub-periods).

It is well known that the NH high latitude winter stratosphere exhibits substantial interannual variability and is influenced by a range of processes and forcings. On the one hand, the relatively long period of the solar cycle can lead to the derived anomalies being affected by aliasing with other atmospheric forcings/processes and/or noise due to interannual variability.

From a modelling perspective, it is therefore crucial that the impact of the solar cycle forcing on climate is studied with sufficiently long simulations.

On the other hand, some other atmospheric forcings and processes (e.g. QBO, ENSO) might influence how the solar-induced anomalies develop and propagate through the stratosphere, e.g. by changing the background state and affecting wave

propagation. In addition, the solar cycle forcing itself might influence various atmospheric forcings and processes. Indications of such non-linear interactions have been found in observations and/or models (see for instance: Salby and Callaghan, 2000; Gray et al., 2004; Pascoe et al., 2005; Labitzke et al., 2006; Haigh and Roscoe, 2006; 2009; Roscoe and Haigh, 2007; Camp and Tung, 2007; Kuroda, 2007; Lu et al., 2009; Calvo and Marsh; 2011, Roy and Haigh, 2011; Matthes et al., 2013). However, given the relatively short length of the satellite record there are still large uncertainties surrounding the derived relationships

(e.g. Anstey and Shepherd, 2013). Clearly, in the case of both observational and modelling studies, analysing such a complex and coupled system using linear techniques, like MLR, could potentially lead to spurious signals upon attempting to decouple individual forcings. Our UM-UKCA results thus suggest the need to focus on not just the solar forcing on its own but also on improving our understanding of the underpinning relationships between the solar cycle and other atmospheric forcings/processes, together with the associated mechanisms.

Lastly, recall that a yearly mean tropospheric warming (up to ~0.1-0.2 K/Wm$^{-2}$) resembling that in the reanalysis was derived in the NH mid-latitudes from the full ensemble (Section 3.2). We suggested that the UM-UKCA anomaly is influenced by the solar cycle signal present in the prescribed observed SSTs/sea-ice, as found to be important in Misios and Schmidt (2013). However, despite the identical SSTs/sea-ice only two members (ENS2 and ENS3) show this mid-latitude tropospheric

warming, while the remaining member (ENS1) shows a warming over the polar region instead. As seen above, there are apparent differences in the wintertime NH high latitude dynamical responses between the individual ensemble members, which are likely to be at least in part caused by the large interannual variability in the region. The results thus suggest that whilst the influence of SSTs/sea-ice appears to enhance the NH mid-latitude tropospheric responses to the solar forcing in models, contributions from the variability in the stratosphere (whether or not solar-induced) and SSTs/sea-ice are important for driving

the responses in the troposphere (in agreement with, e.g., Rind et al., 2008; Meehl et al., 2009; Gray et al., 2016). Given that any set of prescribed SSTs/sea-ice does not necessarily have to be fully consistent with the simulated evolution of the atmosphere, the results indicate the need for coupled atmosphere-ocean models for more confident attribution of tropospheric anomalies to the solar cycle forcing.

**7 Summary**

The 11-year solar cycle is recognised as an important forcing of the climate system. However, there are large uncertainties regarding the signal of solar variability in the atmosphere, which is partly related to uncertainties in the observed response (e.g. Mitchell et al., 2015b; Maycock et al., 2016) as well as marked spread in model-simulated responses (e.g. SPARC CCMVal, 2010; Mitchell et al., 2015a). In this paper, we have presented the first detailed assessment of the atmospheric response to the
11-year solar cycle forcing simulated in the UM-UKCA chemistry-climate model. In contrast to many previous solar cycle studies in the literature, which show solar responses derived using either composite or MLR methodologies, we pay particular attention to the role of detection method by presenting and comparing the results derived using both techniques. In addition, we recognise that interannual variability in the stratosphere can be high, and we examine the impact of the internal atmospheric variability on the derived solar response in UM-UKCA by considering not only the response found from the full three-member
ensemble of 1966-2010 integrations but also the spread of responses found from the individual ensemble members.

Regarding the ensemble mean UM-UKCA response, the enhanced solar cycle activity increases the stratospheric shortwave heating rates and temperatures. The resulting yearly mean warming maximises near the tropical stratopause at ~0.8 K/Wm$^{-2}$. The response occurs at higher altitudes and is ~25% smaller than that derived from ERAI. A number of factors possibly
contributing to this underestimation of model temperature response was identified (Section 3.2.1): i) the relatively broadband shortwave heating scheme; ii) the lack of $O_2$ absorption in the radiation scheme; iii) the underestimation (~20%) of UV (200-320 nm) variability compared with CMIP5; iv) the use of a modest SSI variability (Ermolli et al., 2013); v) uncertainties in the reanalysis (Mitchell et al., 2015b). For ozone, the UM-UKCA model simulates a yearly mean tropical ozone increase of up to ~2.0-2.5 %/Wm$^{-2}$ in the mid-stratosphere. Unlike the more peaked and locally stronger SAGEII ozone response, the
maximum model response is weaker, more horizontally uniform and occurs at lower altitudes. We note that differences exist between the temperature and ozone responses derived from various observational and reanalysis datasets (Soukharev and Hood, 2006; Dhomse et al., 2013; 2016; Mitchell et al., 2015b; Maycock et al., 2016; 2018). Averaged over the globe, the yearly mean total ozone column response simulated in UM-UKCA was estimated (MLR) to be of ~6 DU/Wm$^{-2}$.

The analysis did not find a yearly mean secondary temperature or ozone maximum in the model in the tropical lower stratosphere as seen in the reanalysis. This may be related to differences in the associated dynamical responses in both hemispheres, as manifested by the absence in the model of the yearly mean strengthening of the extratropical stratospheric jets

seen in ERAI. Despite that, we do find a small warming of up to ~0.1-0.2 K/Wm$^{-2}$ in the NH mid-latitude troposphere alongside the associated weakening of the NH subtropical jet on its equatorial side. This tropospheric/lower stratospheric response is in broad qualitative agreement with the reanalysis and suggests a contribution of the solar signal in the prescribed SSTs/sea-ice, as found by Misios and Schmidt (2013).

In accord with the mechanism postulated by Kuroda and Kodera (2002) and Kodera and Kuroda (2002), the enhancement of the horizontal temperature gradient under increased solar cycle activity strengthens and cools the NH stratospheric vortex in autumn. The simulated response extends to the troposphere in November. A sign reversal of the modelled stratospheric response occurs in mid-winter (January). The modulation of the NH polar jet in the model is associated with consistent changes

in planetary wave propagation and, at least in the high latitudes, the meridional overturning circulation. In general, the evolution of the NH dynamical solar response in UM-UKCA during autumn and winter shows some broad resemblance to that seen in ERAI. However, the model shows earlier timing of the responses, which could be related to the positive bias in the model's zonal wind climatology and/or too weak SSI forcing. In addition to the different timing, the simulated westerly response diagnosed from monthly-mean data appears first at higher latitudes than in ERAI, thereby not clearly reproducing

the poleward/downward propagation. In general, any (monthly-mean) westerly anomalies near the subtropical stratopause in UM-UKCA are much weaker and shorter-lived than in ERAI; the UM-UKCA model ensemble also does not reproduce the westerly anomaly observed in the NH mid-/high latitudes in mid-/late spring (April onwards).

Regarding the role of detection method for the derived solar response, we find that the stratospheric solar responses diagnosed

using both the composite and MLR methodologies are, within the associated uncertainty, generally in agreement with each other. Some apparent differences (although mostly not highly statistically significant) are found in the troposphere and in the tropical lower stratosphere. Depending on whether the individual forcings are indeed independent from each other and linear, these could arise either due to noise from the natural/interannual variability and/or aliasing between the solar and other atmospheric forcings, or could in principle be a manifestation of some non-linear interactions between the solar and other

atmospheric forcings and processes. The results highlight that care needs to be taken when investigating the role of the solar cycle forcing in these regions as using only one of the techniques could lead to somewhat different conclusions with regard to the atmospheric impacts of the 11-year solar cycle.

Lastly, in order to understand potential issues in deriving atmospheric responses to the solar cycle forcing from records

comparable in length to the current observational and reanalysis datasets, we discuss the model results derived from the individual ensemble members. We find that the yearly mean tropical temperature and ozone responses derived from the individual integrations in the mid-/upper stratosphere and lower mesosphere are in fair agreement with each other (within the uncertainty) as well as with the response derived from the full ensemble. However, there are larger apparent differences between the individual members in the NH high latitudes; these are mainly related to the apparent differences simulated during

the dynamically active season. The spread of the diagnosed NH responses is particularly large in late autumn/early winter, and gradually lessens later in the season. This suggest that the solar anomalies detected in the highly variable NH high latitudes could be influenced by noise/aliasing due to large variability in the region. In addition, non-linear interactions between the solar and other atmospheric forcings and processes might also play a role over shorter timescales. The UM-UKCA results suggest the need for long timeseries for confident detection of solar anomalies as well as for more research on understanding the relationships between the solar cycle forcing and other atmospheric forcings/processes. Finally, we find that the yearly mean tropospheric warming detected in the NH mid-latitudes from the full ensemble is only reproduced in two out of three ensemble members. This suggests that while the SSTs/sea-ice forcing appears to be an important contributor to the modelled tropospheric responses (Misios and Schmidt, 2013), contributions from both stratospheric variability and SSTs/sea-ice are important for driving the modelled anomalies (see also Rind et al., 2008; Meehl et al., 2009; Gray et al., 2016). The results indicate the need to use coupled atmosphere-ocean models in order to fully capture the impacts of the solar cycle forcing on the tropospheric climate.

**Acknowledgements**

ACM, JAP, PJT and NLA were supported by the National Centre for Atmospheric Science, a NERC funded research centre. We acknowledge funding from the ERC for the ACCI project grant number 267760, including PhD studentship for EMB. ACM acknowledges support from an AXA Postdoctoral Fellowship and NERC Independent Research Fellowship (grant NE/M018199/1).

We acknowledge the use of HECToR, the UK's national high-performance computing service. We thank Markus Kunze for providing the MLR software (including the EESC forcing). The authors also thank Fiona O'Connor and Katja Matthes for providing relevant solar cycle forcing data and/or model code needed for model development.

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

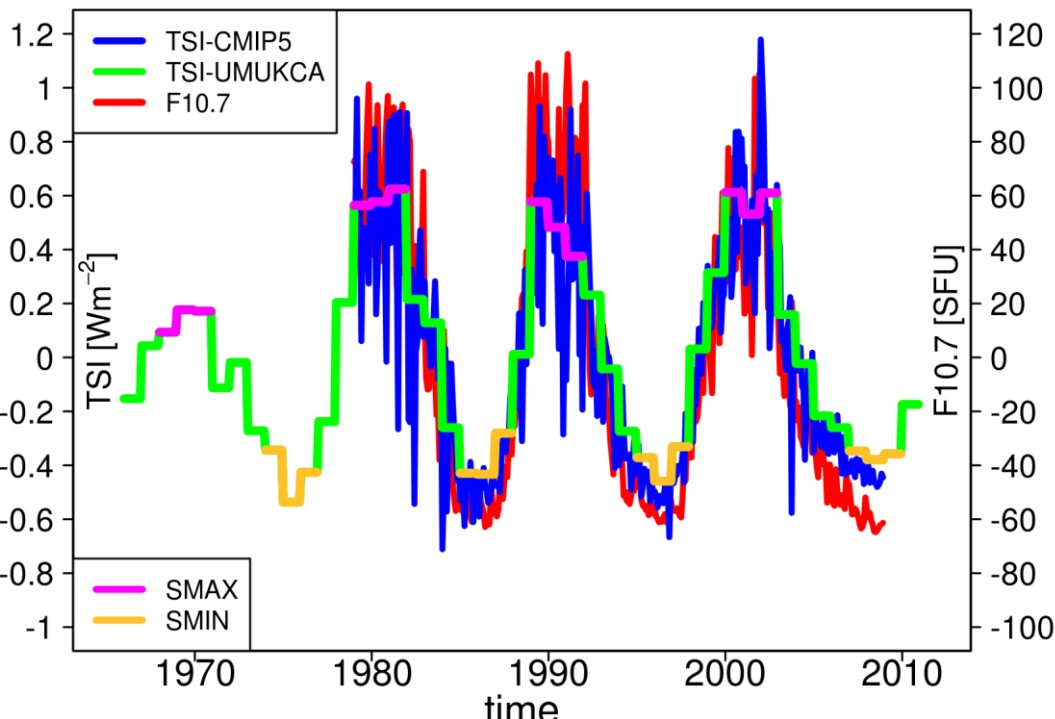

**Figure 1. Timeseries of anomalies in: TSI [Wm$^{-2}$] from the original CMIP5 recommendations (blue; http://solarisheppa.geomar.de/cmip5; Fröhlich and Lean, 1998; Lean, 2000; Wang et al., 2005; Lean, 2009); TSI [Wm$^{-2}$] as given by the processed CMIP5 recommendations imposed in UM-UKCA (green; Fröhlich and Lean, 1998; Lean, 2000; Wang et al., 2005; Lean, 2009; Jones et al., 2011); and the F10.7 cm radio flux [SFU] (red; http://lasp.colorado.edu/lisird/tss/noaa_radio_flux.html). SMAX and SMIN years used in the composite methodology are highlighted in magenta and yellow, respectively.**

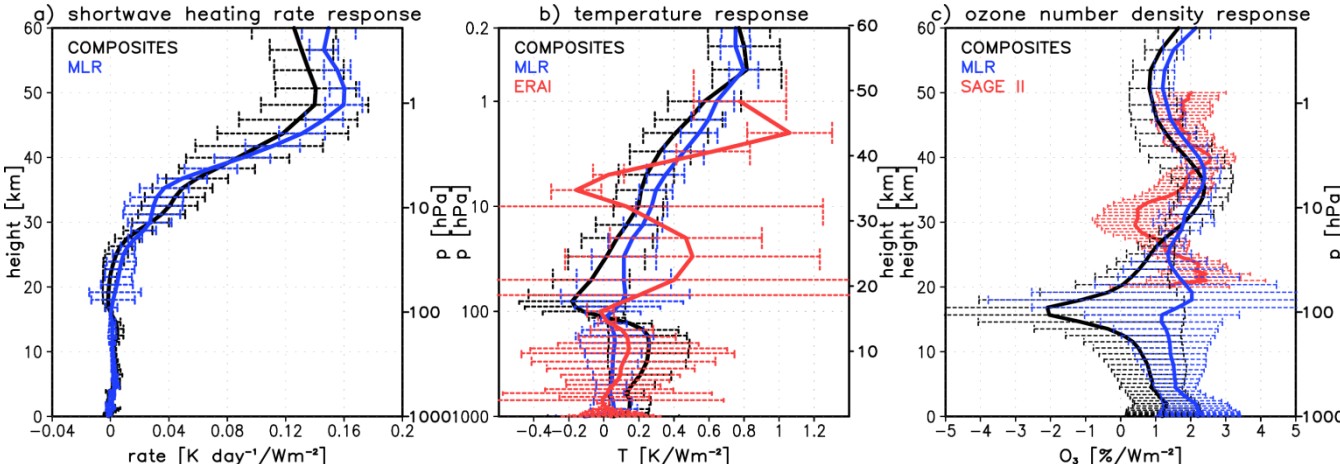

**Figure 2. Yearly mean zonal mean (a) shortwave heating rate [K day$^{-1}$/Wm$^{-2}$], (b) temperature [K/Wm$^{-2}$] and (c) ozone number density [%/Wm$^{-2}$] responses to the 11-year solar cycle forcing in the tropics (25°S-25°N) in UM-UKCA. The UM-UKCA responses were derived using composites (black) and MLR (blue). Shown also in red are the corresponding ERAI MLR temperature (b) and SAGE II MLR ozone (c) responses. The error bars denote the corresponding confidence intervals, represented here by ±2 standard errors of the mean response.**

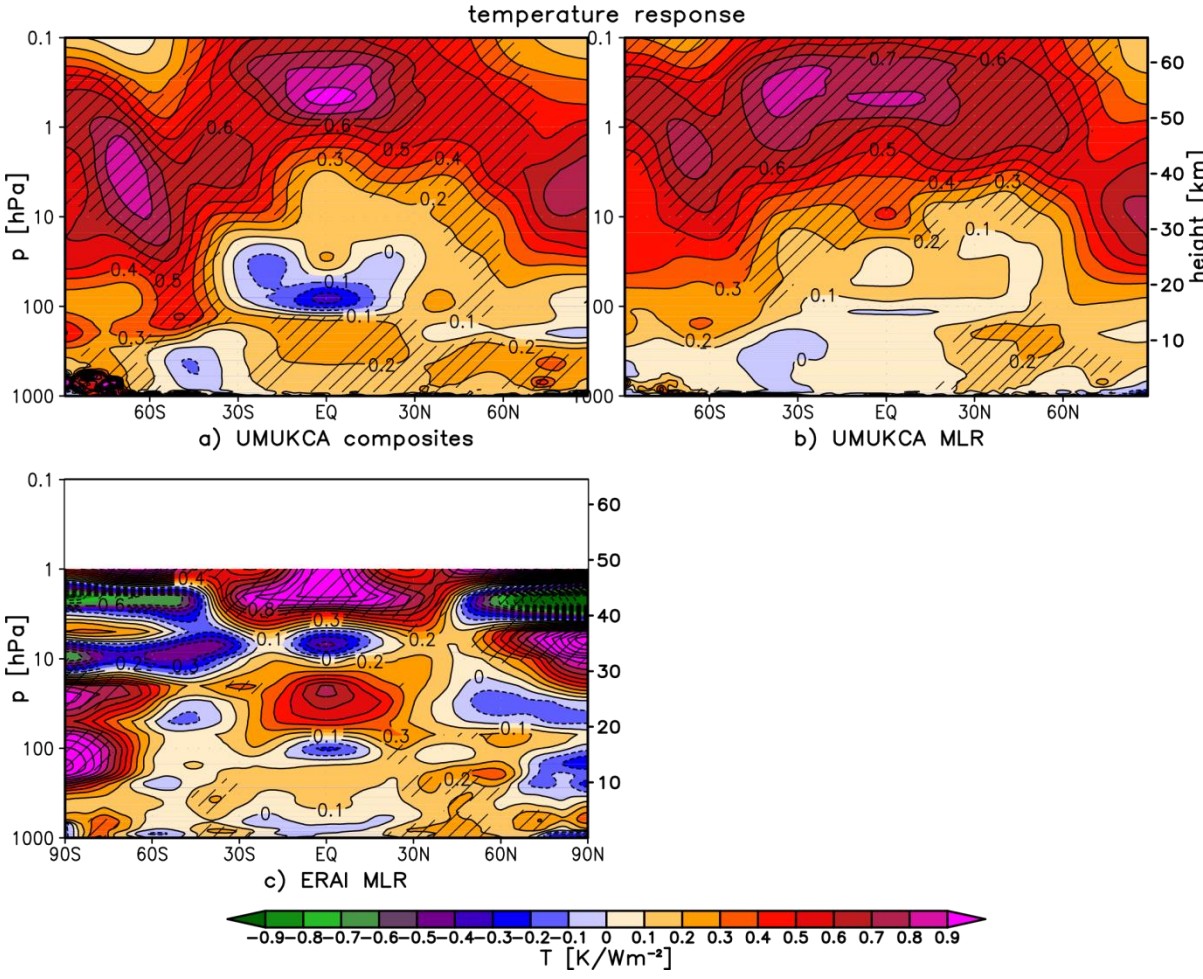

**Figure 3. Yearly mean zonal mean temperature response [K/Wm$^{-2}$] in UM-UKCA derived using composites (a) and MLR (b), as well as in ERAI derived using MLR (c). Shading indicates statistical significance on the 95% level (t-test). Contour spacing is 0.1 K/Wm$^{-2}$.**

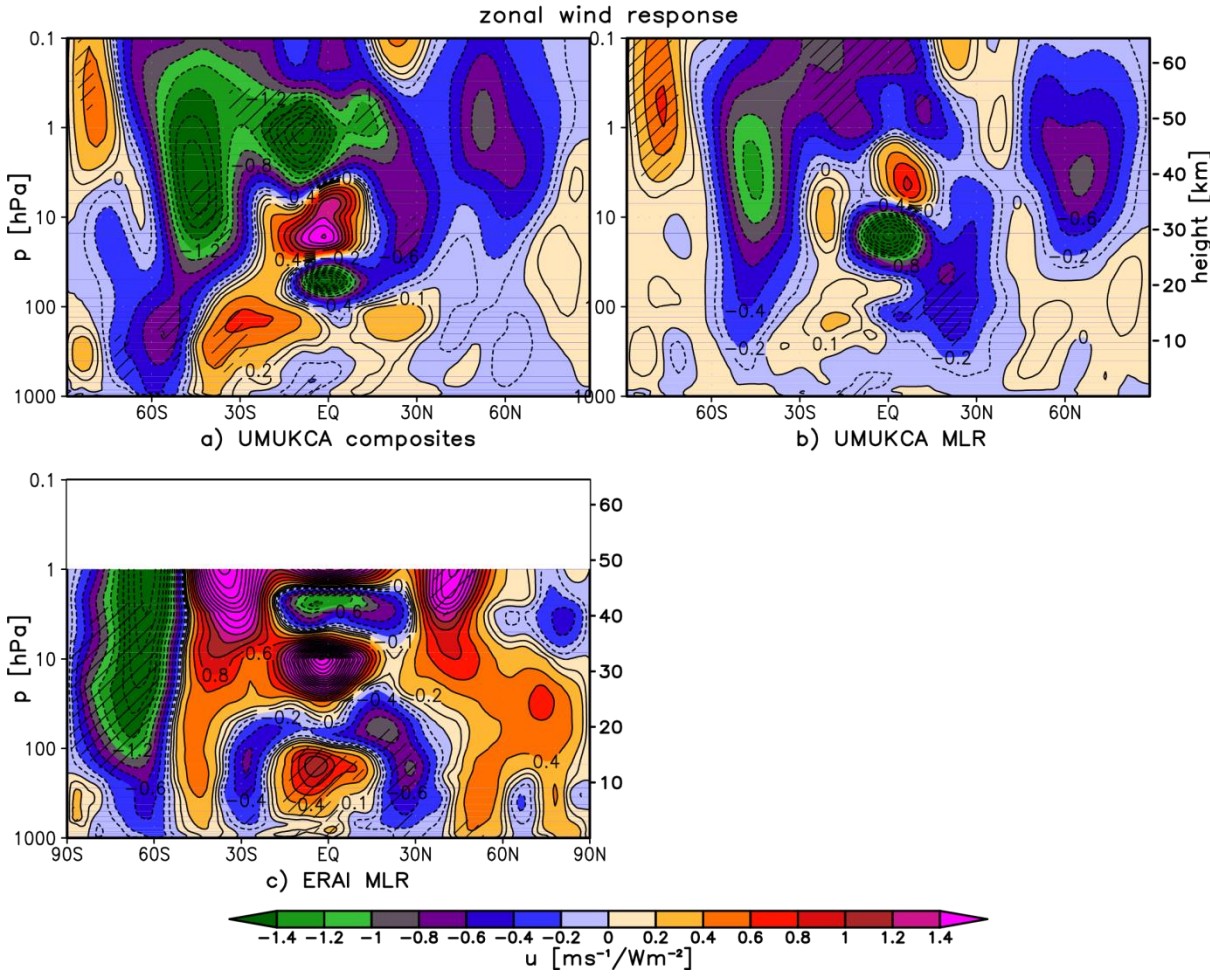

**Figure 4. As in Fig. 3 but for the zonal mean zonal wind response [ms⁻¹/Wm⁻²]. Contour spacing is 0.2 ms⁻¹/Wm⁻²; note also the additional contour at ±0.1 ms⁻¹/Wm⁻².**

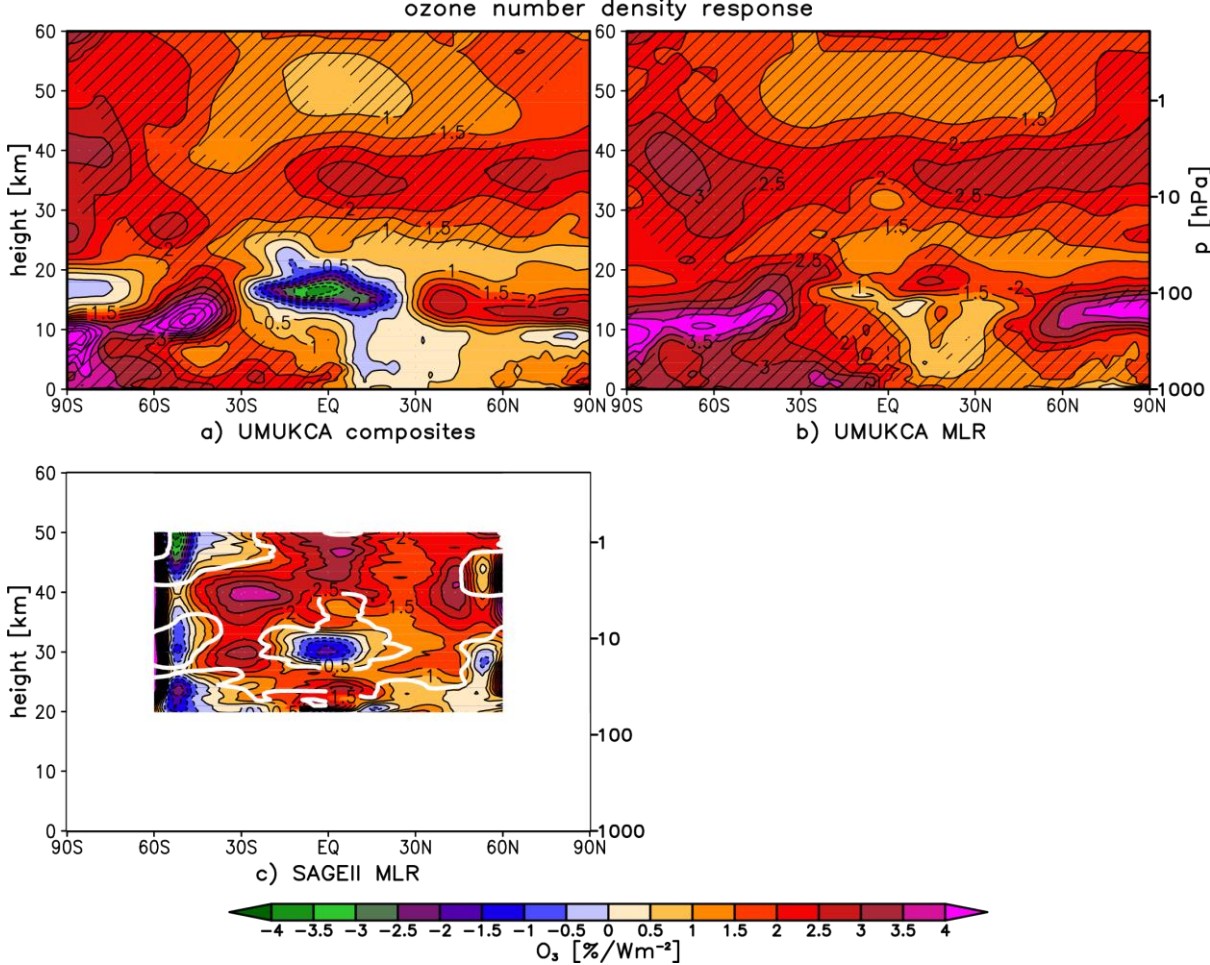

**Figure 5. Yearly mean zonal mean ozone number density response [%/Wm⁻²] in UM-UKCA derived using composites (a) and MLR (b), as we all as in SAGE II derived with MLR (c). The percentages are calculated relative to the mean over the 1966-2010 (a and b) and 1985-2004 (c) periods. Hatching in (a-b) as in Fig. 3. Thick white line in (c) encompasses regions where the statistical significance exceeds the 95% confidence level. Contour spacing is 0.5 %/Wm⁻² up to ± 7 %/Wm⁻².**

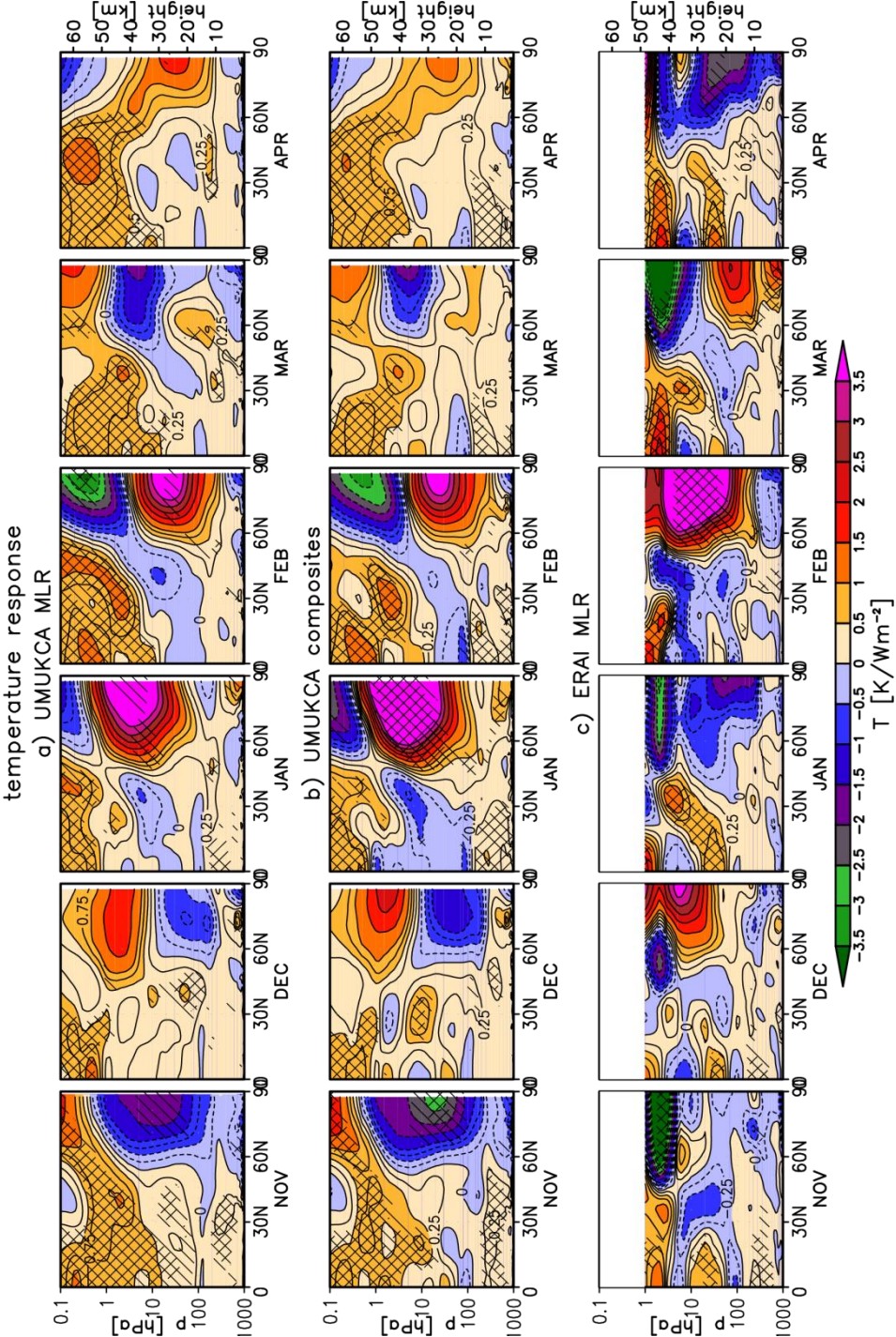

**Figure 6.** Monthly mean November-April zonal mean temperature response [K/Wm⁻²] in UM-UKCA derived using MLR (a) and composites (b). Shown also is the ERAI response derived with MLR (c). Single and double hatching indicates statistical significance on the 90% and 95% level, respectively (t-test). Note the additional contours at ±0.25 K/Wm⁻².

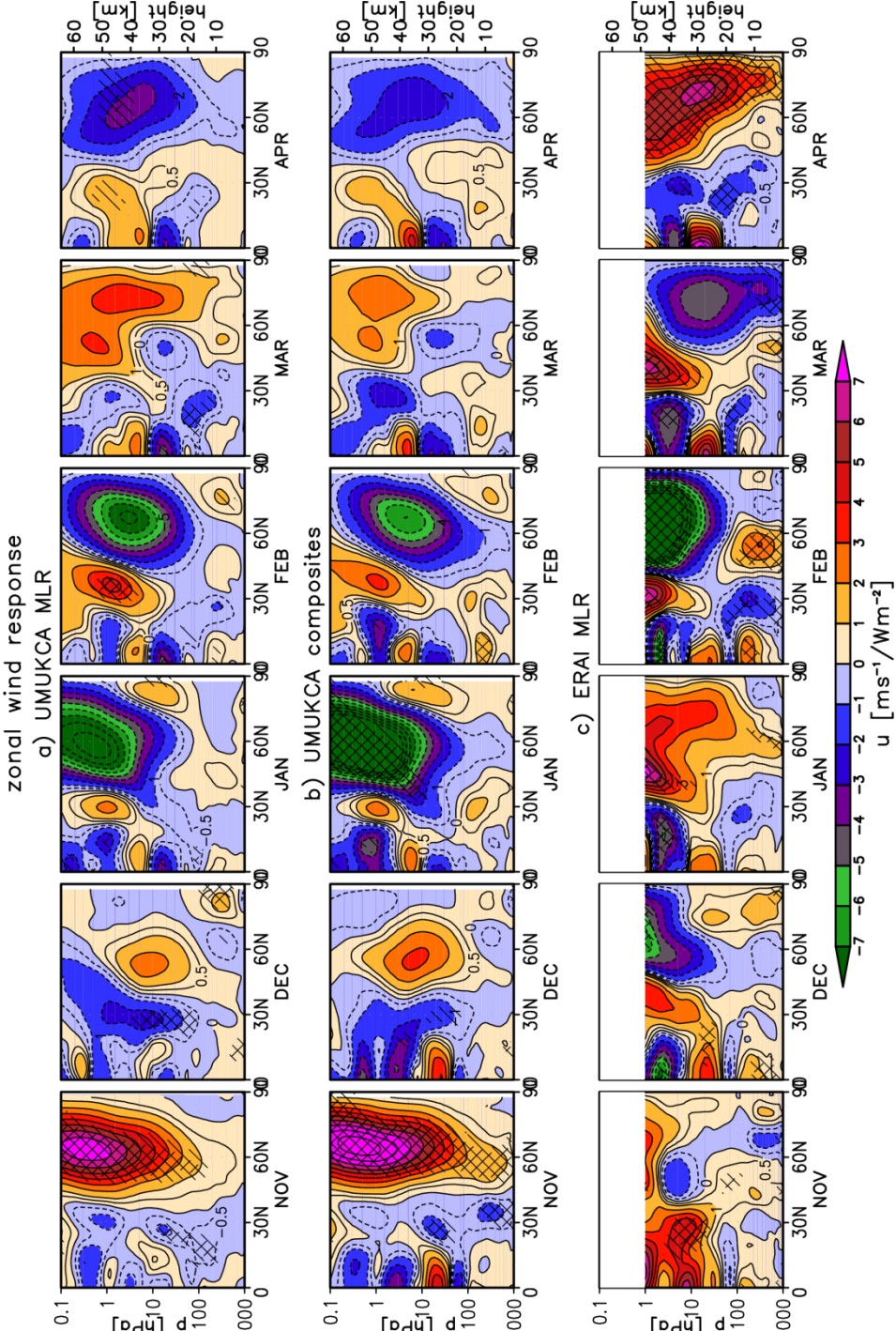

**Figure 7.** As in Fig. 6 but for the zonal mean zonal wind change [ms$^{-1}$/Wm$^{-2}$]. Note the additional contours at ±0.5, ±8, ±9 and ±10 ms$^{-1}$/Wm$^{-2}$.

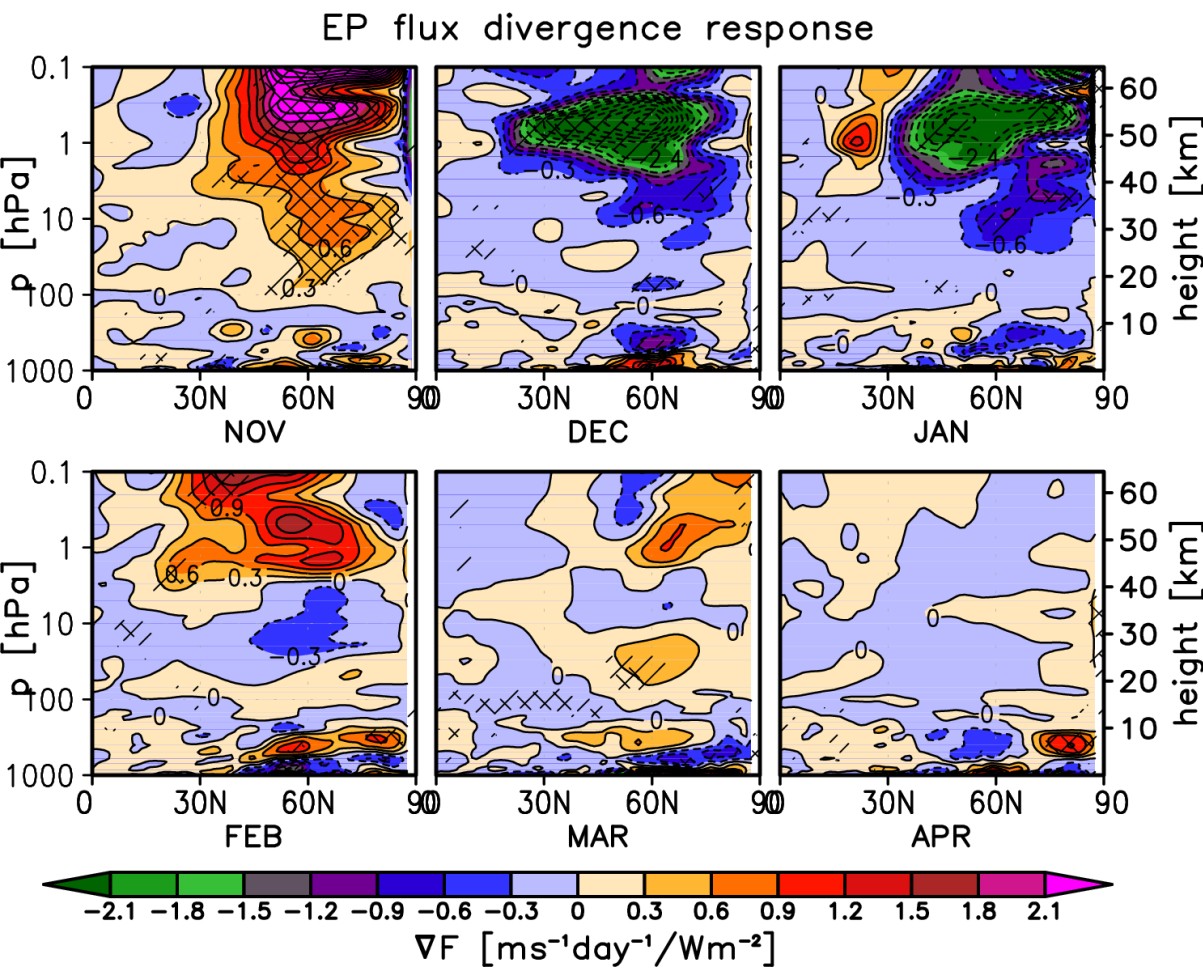

**Figure 8.** Monthly mean November-April scaled[3] EP flux divergence response [ms$^{-1}$day$^{-1}$/Wm$^{-2}$] in UM-UKCA derived with MLR. Single and double hatching indicates statistical significance on the 90% and 95% level, respectively (t-test). Contour spacing is 0.3 ms$^{-1}$day$^{-1}$/Wm$^{-2}$.

---

[3] Scaled EP flux divergence: $(\nabla \cdot \vec{F})/(\rho \cdot a \cdot \cos\phi)$, see Andrews et al. (1987).

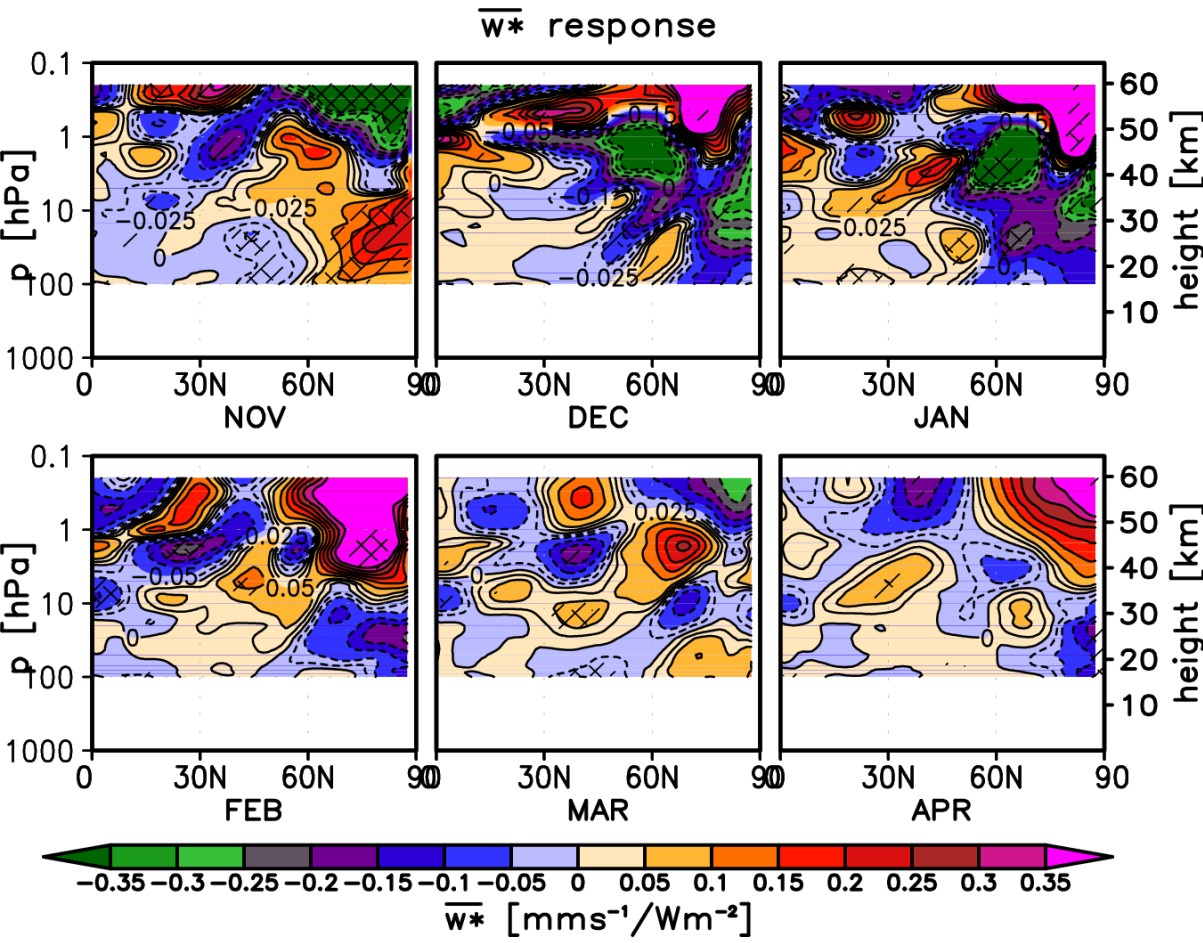

**Figure 9. As in Fig. 8 but for the transformed vertical component of the residual circulation ($\overline{w}^{*}$, see Andrews et al., 1987) response [mms-1/Wm-2] in UM-UKCA. Positive values indicate upwelling. Note the additional contours at ±0.025 mms-1/Wm-2.**

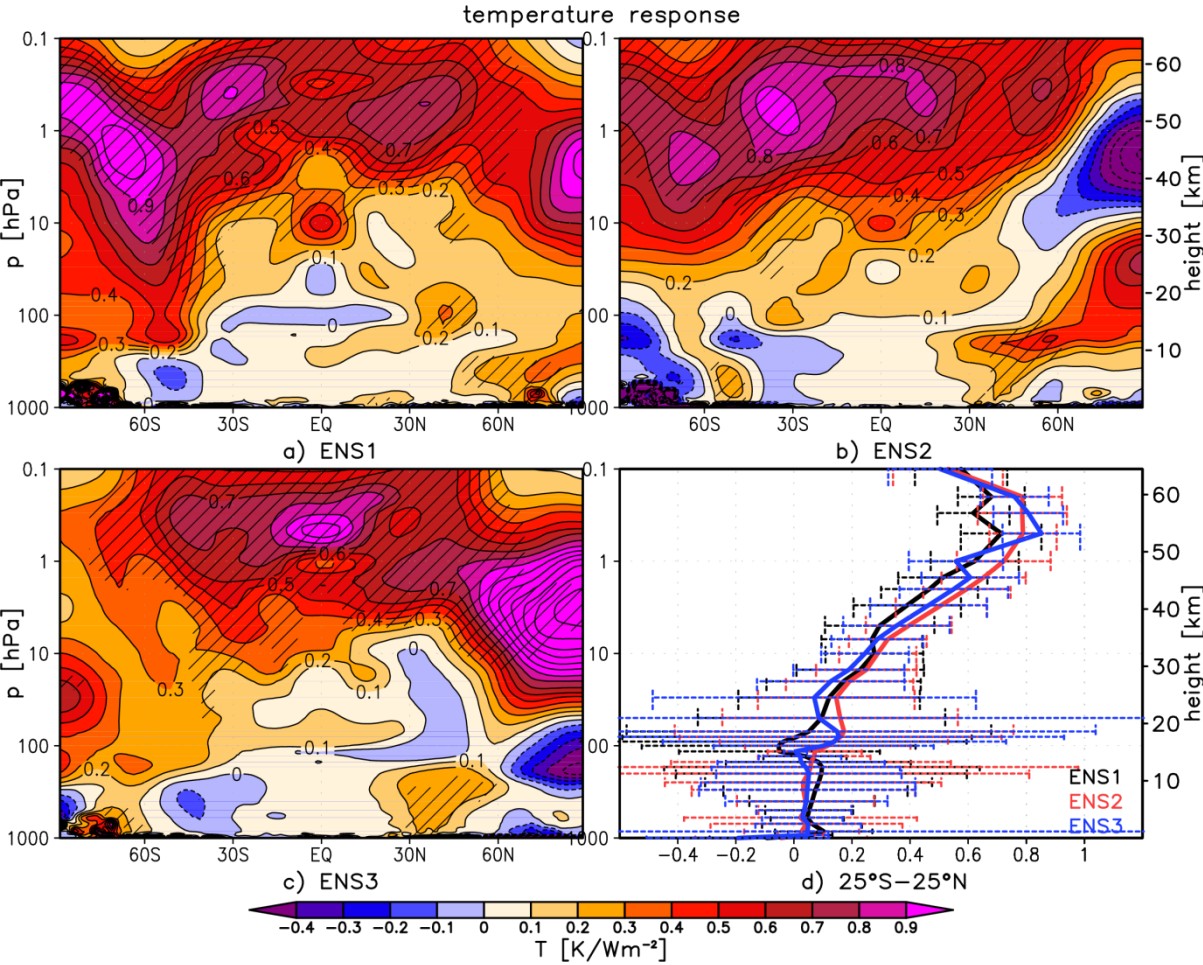

**Figure 10. (a-c)** Yearly mean zonal mean temperature response [K/Wm⁻²] in UM-UKCA derived using MLR for the individual ensemble members (ENS1-3). Hatching indicates statistical significance on the 95% level (t-test). Contour spacing is 0.1 K/Wm⁻². **(d)** The corresponding yearly mean temperature responses in the tropics together with the associated confidence intervals (±2 standard errors).

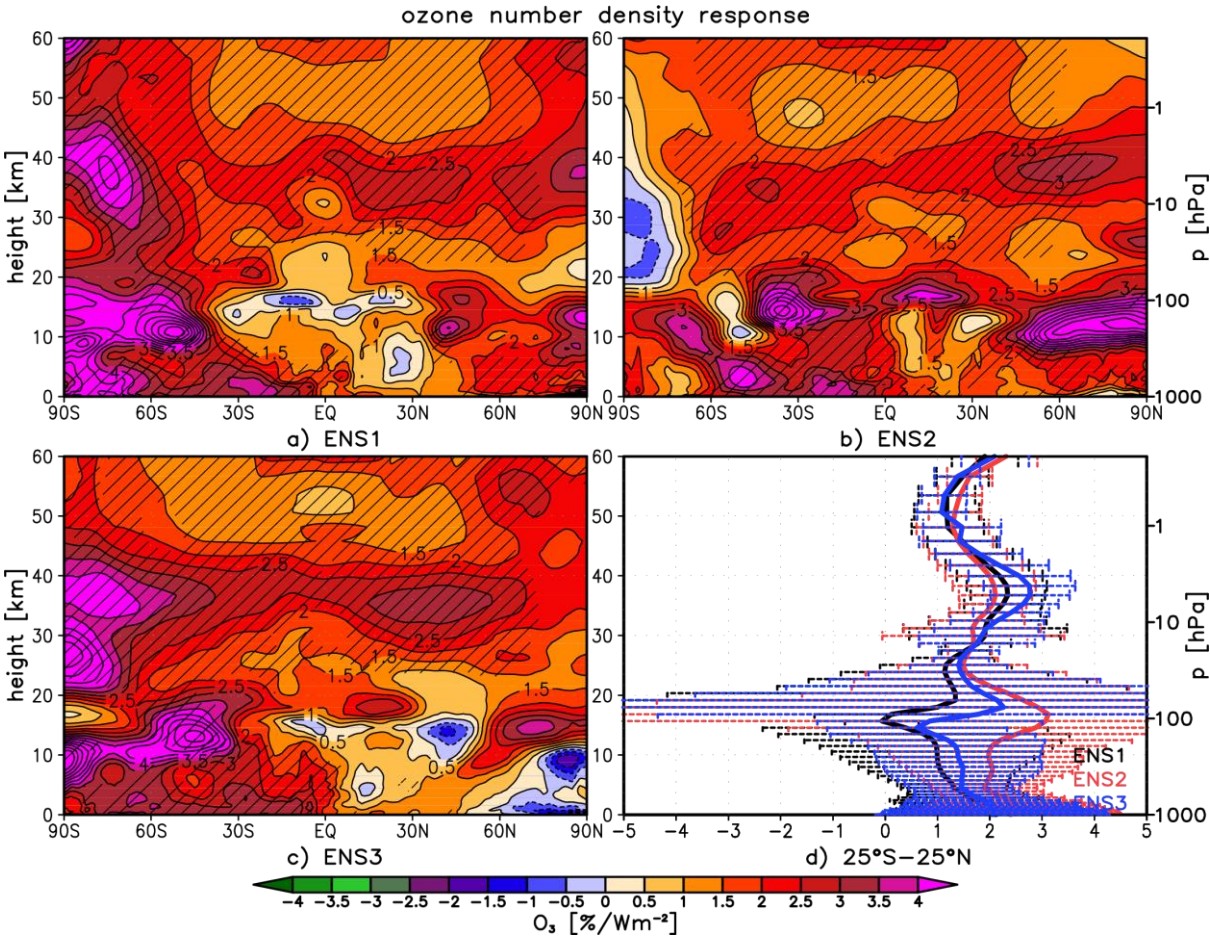

**Figure 11. As in Fig. 10 but for the ozone number density response [%/Wm⁻²]. Contour spacing is 0.5 %/Wm⁻² up to ±7 %/Wm⁻².**

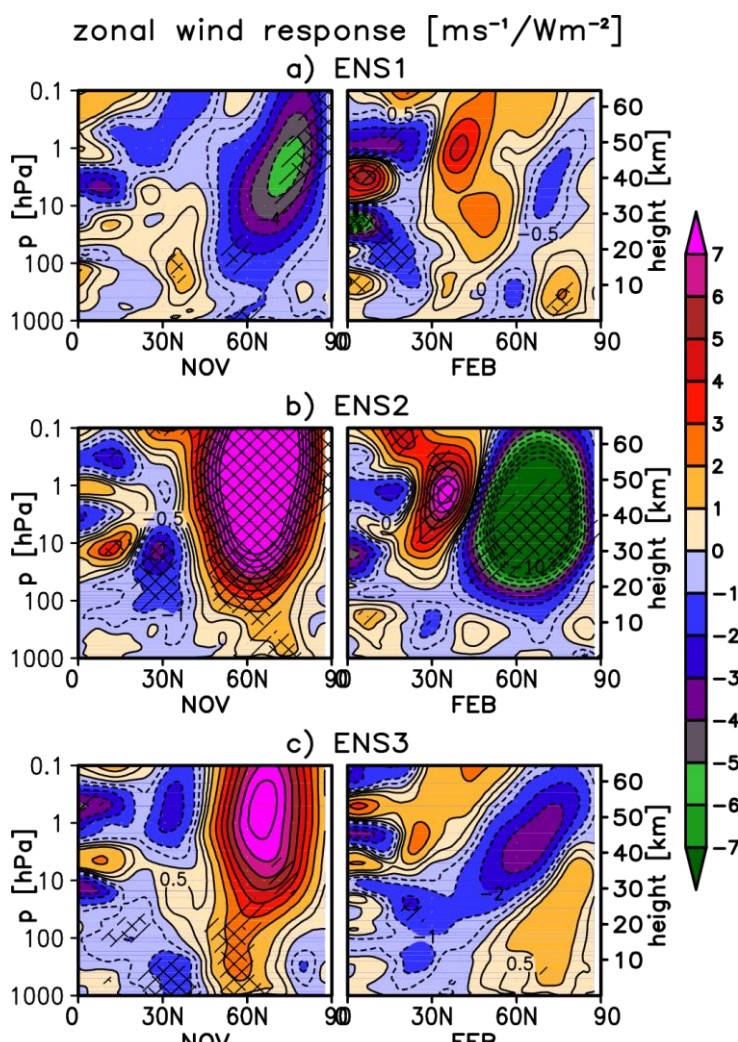

**Figure 12. Monthly mean November (left) and February (right) zonal mean zonal wind response [ms$^{-1}$/Wm$^{-2}$] in UM-UKCA derived using MLR for the individual ensemble members (ENS1-3). Single and double hatching indicates statistical significance on the 90% and 95% level (t-test). Note the additional contours at ±0.5, ±8, ±9 and ±10 ms$^{-1}$/Wm$^{-2}$. See Fig. S5, Supplement, for all months from November to April.**

| Spectral band interval | Absorbing species | SMAX-SMIN irradiance change |
|---|---|---|
| 200 - 320 nm | $O_3$ | 0.16 Wm$^{-2}$ (0.56 %) |
| 320 - 690 nm | $O_3$ | 0.25 Wm$^{-2}$ (0.09 %) |
| 320 - 690 nm | $O_3$, $H_2O$ | 0.20 Wm$^{-2}$ (0.06 %) |
| 0.69 -1.19 µm | $H_2O$, $O_2$ | 0.27 Wm$^{-2}$ (0.06 %) |
| 1.19 - 2.38 µm | $H_2O$, $CO_2$ | 0.15 Wm$^{-2}$ (0.06 %) |
| 2.38 - 10.0 µm | $H_2O$, $CO_2$ | 0.03 Wm$^{-2}$ (0.06 %) |

**Table 1. Yearly mean irradiance change between the years 1981 and 1986 ($\Delta$TSI=1.06 Wm$^{-2}$) for each of the UM-UKCA shortwave heating bands.**

| Latitude | UM-UKCA total ozone column response [DU/Wm$^{-2}$] | |
|---|---|---|
| | Composites | MLR |
| 25°S-25°N | 2.9 (±1.7) | 4.6 (±1.4) |
| 35°N-60°N | 4.8 (±3.3) | 6.1 (±6.5) |
| 35°S-60°S | 6.9 (±3.2) | 8.0 (±3.3) |
| 60°S-60°N | 4.1 (±1.9) | 5.6 (±1.4) |
| 90°S-90°N | 4.2 (±2.2) | 5.9 (±5.4) |

**Table 2. Yearly mean total ozone column response [DU/Wm$^{-2}$], ±2 standard errors, in UM-UKCA derived using composites and MLR.**