# Peer review of "Simulating the atmospheric response to the 11-year solar cycle forcing with the UM-UKCA model: the role of detection method and natural variability"

_Atmospheric Chemistry and Physics, 2018_

## Referee Comment (RC1) · Anonymous Referee #1 · 16 Mar 2018

This is another in a series of model simulation studies of the atmospheric response to 11-year solar forcing. It considers only one model (the UM-UKCA model) but does an extremely thorough and careful statistical analysis of the model simulations (three 45-year simulations) for comparison to a $\sim$ 38-year meteorological reanalysis data set (ERA-Interim). I wish I could say that this study sheds new light on how solar variability influences climate. Unfortunately, as discussed further below, despite the careful analysis of the model and observational data (combined with an excellent review and referencing of previous work), limitations of the model itself, which is representative of

many existing chemistry-climate models, preclude any such progress.

Revisions are requested in response to the following main comments and other more minor comments.

Main comments:

(1) The abstract and summary (section 7) make little mention of the strong disagreements between the model-simulated responses of stratospheric ozone, temperature, and zonal wind to 11-year solar forcing and that derived from observations (here, SAGE II and ERA-Interim). These disagreements (e.g., Figures 3, 4, and 5 for the annual mean response for the 3 simulations combined together) are so strong that there is no chance that the model can simulate realistically the solar-induced climate response. Instead of drawing this obvious conclusion, the abstract and summary (and most of the last half of the manuscript) concentrate on investigating the roles of detection method and internal model variability in affecting (in relatively minor ways) the calculated model response to solar forcing. While the latter aspects are of academic interest, they pale in comparison to the overall failure of the model to simulate the stratospheric response and, hence, any derivative consequences for the troposphere. The reader is left with the false impression that the model does a reasonably good job of simulating the solar response and that any differences with observations can be attributed to uncertainties in reanalysis data sets and possible aliasing of observations by volcanic and ENSO forcing.

(2) One fundamental problem with the model is its failure to adequately simulate the observationally estimated 11-year solar response of the upper stratosphere and lower mesosphere. Such a simulation is essential for initiating a strong zonal wind response near the time of winter solstice, which propagates downward and poleward later in the winter, ultimately leading to a tropospheric response (Kodera and Kuroda, 2002). The ozone response is too weak in the tropical upper stratosphere (dropping to 1 percent

by 45 km; Figure 5a). Other models (see, e.g., Figure 1 of Hood et al., 2015) do better (about 2 percent at 45 km) but still fall short of that estimated from SAGE II data (about 3 percent). This shortfall of the simulated ozone response is at least partly responsible for the too-weak model temperature and zonal wind responses near the stratopause evident in Figures 3 and 4. At least some other climate models do a much better job of simulating the temperature response at these altitudes (see, e.g., Figures 4 and 5 of Mitchell et al., QJRMS, 2015) but I did not notice any mention of this in the text. Possible reasons for the poor ozone simulation are discussed only briefly in section 3.3.2 on p. 12. One factor that is not mentioned is the very coarse resolution of the 6 spectral bands evident in Table 1. The first spectral interval includes the entire UV region from 200 to 320 nm. As mentioned at the bottom of p. 9, the broad spectral band in the UV also has negative effects on the model's shortwave heating scheme and therefore its ability to simulate the full magnitude of the temperature response in the upper stratosphere and lower mesosphere. All of these problems combined together inevitably prevent the model from simulating realistically the downward propagating solar-induced dynamical signal. It is these deficiencies that should be emphasized in the abstract and summary sections for a fair assessment.

More minor comments:

(3) The model produces essentially no tropical lower stratospheric response of ozone and temperature, which contrasts with that derived from SAGE II and ERA-Interim data (Figures 3 and 5). This is probably mainly because of the weak modeled upper stratospheric response, which leads to, at most, a weak perturbation of the tropical up-welling rate via the Brewer-Dobson circulation. However, it may also be partly because the model does not have a coupled ocean; it only uses prescribed sea surface temperatures, etc. Recent work shows that there is strong coupling between the tropical lower stratosphere and troposphere, which will not be adequately simulated in a model

with no coupled ocean. For example, observational analyses have recently shown that the stratospheric QBO influences tropical convection and the Madden-Julian oscillation (Yoo and Son, GRL, 2016; and references therein). If the same is true for solar forcing, positive feedbacks from the tropical tropospheric response may have the effect of amplifying the tropical lower stratospheric response in ways that would not be simulated in a model with no coupled ocean and no MJO. Please at least mention with references in the revised manuscript introduction this new evidence for coupling between the tropical lower stratosphere and tropical tropospheric convection.

(4) With regard to the above noted evidence for an influence of the stratospheric QBO on the tropical troposphere, have the authors investigated whether such an influence can be simulated in the UM-UKCA model? Shouldn't such an investigation come first since many more QBO cycles are available for a robust statistical analysis? Presumably the model cannot simulate this since it has no coupled ocean and therefore, probably, no MJO. If not, then some discussion should be added to the manuscript about this deficiency of the model and whether it could affect the model's ability to simulate as well the 11-year solar forcing. Also, please mention in section 2.1.1 that the model is not able to simulate the MJO if this is the case.

(5) The font chosen for printing most of the manuscript text is difficult to read.

(6) P. 5, section 2.1.2. Is the resolution of the model's SSI forcing daily or monthly? This is not clear from Figure 1 after 1979 where large fluctuations are present near solar maximum. In other words, does it only simulate the 11-year component of SSI variability or does it also simulate the 27-day component? The latter could have some non-negligible effects on the simulated 11-year atmospheric response since extrema of the 27-day cycle can differ significantly from the mean. While it is true that TSI and UV proxies such as F10.7 correlate well on the 11-year time scale as shown in Figure 1 (see text on p. 8, line 18), this is not at all true on the 27-day time scale. Also, F10.7

does not correlate adequately with the actual solar flux near 200 nm on the 27-day time scale. So, if daily resolution is used in the future, you would need to use the actual SSI near 200 nm as your solar variable.

(7) The monthly analyses for the Nov. to April season in section 4 are useful for showing that the model only simulates a strong zonal wind response in November centred near 60N latitude whereas observations indicate that the zonal wind response is initiated at much lower latitudes in the subtropics and continues on with downward and poleward propagation through the winter. The same high-latitude bias of the 11-year zonal wind response is seen in most or all climate models. No clear explanation for this bias has been advanced but it could be related to the ozone response, which is much larger in the SAGE II observations in the tropical upper stratosphere than it is in most models.

(8) Most of the manuscript beginning with section 4.2 could be criticized as being either over-analysis of a deficient model or an investigation of issues that are mainly of academic interest. Can the authors find ways to delete or at least shorten some of this material?

---

## Referee Comment (RC2) · Anonymous Referee #2 · 23 Mar 2018

This is an interesting and very well written study on solar 11-year signatures in different atmospheric parameters (in the troposphere, stratosphere and lower thermosphere) based on model simulations with the UM-UKCA model. The paper does not present any spectacular new results on atmospheric effects of solar variability at the 11-year scale, but it is an interesting contribution to the field and should eventually be published in my opinion. An important aspect of the study is the fact that two different analysis techniques (i.e., a composite analysis and multi-linear regression) are applied and the differences in the results are studied and discussed. I ask the authors to consider the

following comments:

Page 2, line 15: "The spectral distribution of solar irradiance is commonly referred to as the spectral solar irradiance (SSI)."

I disagree, this is not the correct meaning of solar spectral irradiance. SSI has the units W / m2 / nm, i.e. spectral irradiance. It is the power of electromagnetic radiation per unit area and per spectral interval. SSI at a certain wavelength can also be determined or calculated without considering the spectral distribution of the entire spectrum.

Page 5, line 29: "In the Fast-JX photolysis scheme, the change in partitioning of solar irradiance . . ."

I find the phase "change in partitioning" a little misleading, because it's not only the partitioning that's changing. The overall TSI changes as well.

Page 5, last line: "At pressures less than 0.2 hPa, i.e. where photolysis rates are calculated using the look up tables, the 11-year solar cycle variability is reflected in the TSI change only, with no modulation of SSI."

I don't really understand this statement. If TSI is changed, then SSI (in a given spectral interval) must change as well. You probably mean that the spectral distribution of the solar irradiance spectrum is not changed, right? See also my comment on the meaning of SSI above.

Page 6, line 22: ".. but a sparse horizontal sampling (Soukharev and Hood, 2006; Hood et al., 2015; Tummon et al., 2015)."

I think it's more appropriate here to cite one of the original instrument or algorithm papers, rather than papers that "only" use the data. Sparse geographical coverage was always known to be a disadvantage of solar occultation observations.

Page 7, equation 5 (and the equation in the supplement): The choice of the offset and trend terms does not make sense to me. The offset is just a number, right? Why does

it have to be represented by a product of two numbers. This is not necessary and only makes things more complicated. I doubt that the function is implemented in this way in your fitting routine – this would not lead to stable results. Also, the trend term trend(t) is simply "t", right? If yes, then it should be written that way.

Page 7, line 16: "(here applied 5 times)"

Is there a specific reason, why this filter was applied 5 times?

Page 8, line 14: "However, unlike the yearly mean TSI timseries that forces the model, the timeseries chosen here is that originally recommended for the CMIP5 models"

How does this choice affect the results? Ideally, the same solar proxy time series should be used. Please add a brief (qualitative) comment on the expected impact (probably very small).

Same line: "timseries" -> "timeseries"

Section 3: It would be good to show a sample result of the MLR analysis (fit and residual). I have no reason to doubt that the method works well, but it's always good to see a fit example.

Page 10, line 17: "According to the postulated . . ."

I think this sentence is incomplete.

Page 11, section 3.2.3: This section focuses more on the (few) similarities between ERAI and the model simulations. However, looking at Figs. 3 and 4 the obvious aspects are the significant differences for both T and the zonal wind response. They should be mentioned/discussed as well.

Page 12, line 14: I suggest replacing "The lower altitude of the ozone response" by "The lower altitude of the maximum ozone response"

Page 17, line 10: "we find that the total column ozone responses derived in various

regions are somewhat higher for MLR than for composites,"

Any ideas on the causes of this behaviour?

Page 17, line 28: "observational records such as ERAI"

Can one really call ERAI an observational record? It's certainly different from the "pure" observational records such as the SAGE II O3 data set.

Page 21, line 18: "Some differences (although not statistically significant) are found in the troposphere and in the tropical lower stratosphere."

Did the paper really show that the differences are not statistically significant? Some signatures are statistically significant in one analysis, but not in the other. What does this imply in terms of the statistical significance of the differences?

Figure 2, caption and title of panel a): "heating rates response" -> "heating rate response"

Page 43, Table 1, lines 3 and 4: Both lines list the same spectral interval (320 – 690 nm). Is this intended? If yes, the exact meaning of these two lines (and their difference) is not clear to me.

---

## Author Response (AR1)

Response to Referee #1

**This is another in a series of model simulation studies of the atmospheric response to 11-year solar forcing. It considers only one model (the UM-UKCA model) but does an extremely thorough and careful statistical analysis of the model simulations (three 45-year simulations) for comparison to a ~ 38-year meteorological reanalysis data set (ERA-Interim). I wish I could say that this study sheds new light on how solar variability influences climate. Unfortunately, as discussed further below, despite the careful analysis of the model and observational data (combined with an excellent review and referencing of previous work), limitations of the model itself, which is representative of many existing chemistry-climate models, preclude any such progress. Revisions are requested in response to the following main comments and other more minor comments.**

We thank the reviewer for their constructive comments. We reply to the specific comments below in blue.

We appreciate the reviewer's comments regarding the challenges of modelling the effects of the solar cycle variability on the atmosphere. However, we would like to stress that our primary motivation is to demonstrate the aspects of the solar cycle response that are robust and less robust to analysis methodology, thereby aiding interpretation of the solar cycle signals found in observational and modelling datasets. We use the new capability in UM-UKCA as a basis where careful analysis and testing is possible. One point to note in relation to some of the reviewer's comments is the large intrinsic uncertainty in the diagnosed signal of the 11-year solar cycle in observations/reanalysis, which can be understood from our study as being partly related to the large internal variability in the stratosphere, and partly related to issues associated with the datasets themselves (e.g. Mitchell et al., 2015b; Maycock et al., 2016). Hence, if the modelled best estimate response to solar forcing differs from the observations/reanalysis best estimate, this does not automatically mean the model is wrong as the reviewer suggests, as we must also consider the uncertainty associated with the two estimates.

**Main comments:**

**(1) The abstract and summary (section 7) make little mention of the strong disagreements between the model-simulated responses of stratospheric ozone, temperature, and zonal wind to 11-year solar forcing and that derived from observations (here, SAGE II and ERA-Interim). These disagreements (e.g., Figures 3, 4, and 5 for the annual mean response for the 3 simulations combined together) are so strong that there is no chance that the model can simulate realistically the solar-induced climate response. Instead of drawing this obvious conclusion, the abstract and summary (and most of the last half of the manuscript) concentrate on investigating the roles of detection method and internal model variability in affecting (in relatively minor ways) the calculated model response to solar forcing. While the latter aspects are of academic interest, they pale in comparison to the overall failure of the model to simulate the stratospheric response and, hence, any derivative consequences for the troposphere. The reader is left with the false impression that the model does a reasonably good job of simulating the solar response and that any differences with observations can be attributed to uncertainties in reanalysis data sets and possible aliasing of observations by volcanic and ENSO forcing.**

We first respond to the reviewer's assertion that there is a strong disagreement between the solar cycle signals diagnosed in UM-UKCA and observations/reanalysis. For example, in Fig. 2. the uncertainties in the best estimates of temperature and ozone responses in the tropics are overlapping throughout most of the stratosphere. The main exception to this is for temperature in the tropical upper stratosphere, where ERA-Interim shows a stronger peak temperature change. However, this is not found consistently across reanalysis datasets (Mitchell et al., 2015b), and it is known that uncertainties exists in reanalysis datasets in the upper stratosphere (and above) due to scarcity of long-term measurements (e.g. McLandress et al., 2014; Mitchell et al., 2015a; 2015b; Long et al., 2017). Thus, while we must of course make use of the available observational/reanalysis datasets, it should be recognised that these provide relatively weak constraints for models owing to the large uncertainty in the diagnosed solar cycle signals. This is the case for the ozone response (e.g. Dhomse et al., 2013; 2016; Maycock et al. 2016) and for the temperature and zonal wind response (Mitchell et al., 2015b). We therefore believe that our results in Sections 5-6 provide a valuable insight into the role of detection method and interannual variability for the detected solar cycle signal/response that has not been widely acknowledged in the previous literature.

We also wish to respond to the reviewer's point that "investigating the roles of detection method and internal model variability in affecting (in relatively minor ways) the calculated model response to solar forcing." While we show that the choice of analysis method has only a small effect on the annual mean ozone and temperature responses in the stratosphere, which is a useful result in itself, at high northern latitudes in winter the differences between the individual ensemble members are as large, or larger than, the differences between the ensemble mean response and ERAI-Interim. This finding alone emphasises the potential for internal atmospheric variability to affect solar signal detection in a system with relatively few degrees of freedom (i.e. covering only a few solar cycles).

Despite the above points, we accept that there are some differences between the solar signals in UM-UKCA and observational/reanalysis datasets which we did not try to hide. As suggested by the reviewer, we make it even clearer by adding the following paragraph to the abstract: "While many of the diagnosed signals of the solar cycle in UM-UKCA agree with those derived from observations/reanalysis within their respective uncertainties, there are some differences in magnitude, spatial structure and timing of the signals in ozone, temperature and zonal winds. This could be due to the large uncertainties in the observational estimates of the solar response and/or deficiencies in the model performance".

**(2) One fundamental problem with the model is its failure to adequately simulate the observationally estimated 11-year solar response of the upper stratosphere and lower mesosphere. Such a simulation is essential for initiating a strong zonal wind response near the time of winter solstice, which propagates downward and poleward later in the winter, ultimately leading to a tropospheric response (Kodera and Kuroda, 2002). The ozone response is too weak in the tropical upper stratosphere (dropping to 1 percent by 45 km; Figure 5a). Other models (see, e.g., Figure 1 of Hood et al., 2015) do better (about 2 percent at 45 km) but still fall short of that estimated from SAGE II data (about 3 percent). This shortfall of the simulated ozone response is at least partly responsible for the too-weak model temperature and zonal wind responses near the stratopause evident in Figures 3 and 4. At least some other climate models do a much better job of simulating the temperature response at these altitudes (see, e.g., Figures 4 and 5 of Mitchell et al., QJRMS, 2015) but I did not notice any mention of this in the text. Possible reasons for the poor ozone simulation are discussed only briefly in section 3.3.2 on p. 12. One factor that is not mentioned is the very coarse resolution of the 6 spectral bands evident in Table 1. The first spectral interval includes the entire UV region from 200 to 320 nm. As mentioned at the bottom of p. 9, the broad spectral band in the UV also has negative effects on the model's shortwave heating scheme and therefore its ability to simulate the full magnitude of the temperature response in the upper stratosphere and lower mesosphere. All of these problems combined together inevitably prevent the model from simulating realistically the downward propagating solar-induced dynamical signal. It is these deficiencies that should be emphasized in the abstract and summary sections for a fair assessment.**

We first address the reviewer's comment: "The ozone response is too weak in the tropical upper stratosphere (dropping to 1 percent by 45 km; Figure 5a). Other models (see, e.g., Figure 1 of Hood et al., 2015) do better (about 2 percent at 45 km) but still fall short of that estimated from SAGE II data (about 3 percent)." We begin noting that while the reviewer's comment makes reference to the composite model results in Fig. 5a they do not mention that the corresponding MLR ozone response (Fig. 5b) at ~45 km is larger (~1.5%) and statistically significant than the composites. This indicates the MLR method is better suited for extracting the ozone response in this region. The apparent difference between the model and satellite observed best-estimate solar-ozone response is therefore smaller than suggested by the reviewer. We note that we have, however, already mentioned the differences between the model and SAGEII in the manuscript (see Section 3.3.2.). We would like to emphasise again here that there are large uncertainties in the observed solar-ozone response in the tropical upper stratosphere, and that the uncertainties in the model and satellite tropical ozone signals are overlapping (see Fig. 2c), which precludes the conclusion that the model ozone response is too weak.

Secondly we address the reviewer's comment: "Possible reasons for the poor ozone simulation are discussed only briefly in section 3.3.2 on p. 12. One factor that is not mentioned is the very coarse resolution of the 6 spectral bands evident in Table 1." We apologise for this confusion: the 6 spectral bands applies only for the model shortwave radiative transfer scheme (discussed further below). The Fast-JX photolysis scheme from UM-UKCA consists of a total of 18 bands over the spectral region 177-850 nm (see Section 2 of the manuscript). The ability of this 18-band version of Fast-JX to simulate the solar-ozone response was evaluated by Sukhodolov et al. (2016) against detailed line-by-line calculations. As noted in our manuscript, "Sukhodolov et al. (2016) found a small (~0.2-0.3 %) underestimation of the ozone response to the solar cycle in the upper stratosphere in the Fast-JX code used in UM-UKCA compared with line-by-line calculations; this gave rise to only a small temperature underestimation of ~0.05 K". Hence the resolution of the photolysis scheme is unlikely to be a major contributor to any differences between the modelled and observed ozone response.

**More minor comments:**

**(3) The model produces essentially no tropical lower stratospheric response of ozone and temperature, which contrasts with that derived from SAGE II and ERA-Interim data (Figures 3 and 5). This is probably mainly because of the weak modeled upper stratospheric response, which leads to, at most, a weak perturbation of the tropical upwelling rate via the Brewer-Dobson circulation. However, it may also be partly because the model does not have a coupled ocean; it only uses prescribed sea surface temperatures, etc. Recent work shows that there is strong coupling between the tropical lower stratosphere and troposphere, which will not be adequately simulated in a model with no coupled ocean. For example, observational analyses have recently shown that the stratospheric QBO influences tropical convection and the Madden-Julian oscillation (Yoo and Son, GRL, 2016; and references therein). If the same is true for solar forcing, positive feedbacks from the tropical tropospheric response may have the effect of amplifying the tropical lower stratospheric response in ways that would not be simulated in a model with no coupled ocean and no MJO. Please at least mention with references in the revised manuscript introduction this new evidence for coupling between the tropical lower stratosphere and tropical tropospheric convection.**

We agree with the reviewer that feedbacks from the ocean can be important for the atmospheric response to the solar cycle. However, we note that our simulations are forced with prescribed SSTs from observations (HadISST, Rayner et al., 2003) and therefore implicitly include a solar cycle signal which may contribute to the simulated atmospheric response. This is explained in Section 3.2.3. To address the reviewer's request, we have added a sentence to the introduction regarding the possible role of atmosphere-ocean coupling for the tropical lower stratospheric response.

**(4) With regard to the above noted evidence for an influence of the stratospheric QBO on the tropical troposphere, have the authors investigated whether such an influence can be simulated in the UM-UKCA model? Shouldn't such an investigation come first since many more QBO cycles are available for a robust statistical analysis? Presumably the model cannot simulate this since it has no coupled ocean and therefore, probably, no MJO. If not, then some discussion should be added to the manuscript about this deficiency of the model and whether it could affect the model's ability to simulate as well the 11-year solar forcing. Also, please mention in section 2.1.1 that the model is not able to simulate the MJO if this is the case.**

See the response to (3).

**(5) The font chosen for printing most of the manuscript text is difficult to read.**

We are sorry to hear that the reviewer found this to be the case but, unfortunately, we had to use the required ACPD template for manuscript production. Therefore, the choice of font there was outside of our control.

**(6) P. 5, section 2.1.2. Is the resolution of the model's SSI forcing daily or monthly? This is not clear from Figure 1 after 1979 where large fluctuations are present near solar maximum. In other words, does it only simulate the 11-year component of SSI variability or does it also simulate the 27-day component? The latter could have some non-negligible effects on the simulated 11-year atmospheric response since extrema of the 27-day cycle can differ significantly from the mean. While it is true that TSI and UV proxies such as F10.7 correlate well on the 11-year time scale as shown in Figure 1 (see text on p. 8, line 18), this is not at all true on the 27-day time scale. Also, F10.7 does not correlate adequately with the actual solar flux near 200 nm on the 27-day time scale. So, if daily resolution is used in the future, you would need to use the actual SSI near 200 nm as your solar variable.**

The model is forced with yearly mean SSI (as stated in Section 2.1.2) and, therefore, only simulates the 11-year component of SSI variability.

**(7) The monthly analyses for the Nov. to April season in section 4 are useful for showing that the model only simulates a strong zonal wind response in November centred near 60N latitude whereas observations indicate that the zonal wind response is initiated at much lower latitudes in the subtropics and continues on with downward and poleward propagation through the winter. The same high-latitude bias of the 11-year zonal wind response is seen in most or all climate models. No clear explanation for this bias has been advanced but it could be related to the ozone response, which is much larger in the SAGE II observations in the tropical upper stratosphere than it is in most models.**

We first note that much of the downward and poleward propagation of anomalies found in reanalysis is not highly statistically significant (Mitchell et al., 2015b). Accordingly, as illustrated in Fig. R1 below, the errorbars associated with the NH winter UM-UKCA and ERAI zonal wind responses largely overlap:

[Figure]

Fig. R1. November to April zonal mean zonal wind response [ms$^{-1}$/Wm$^{-2}$] at 60ºN diagnosed using MLR for UM-UKCA ensemble (blue) and ERA-I (red). The errorbars denote ±2 standard errors of the mean response.

We further note that the UM-UKCA simulations in Fig. 12 are 45 years long, which is longer than modern reanalyses covering the satellite era. Yet the individual ensemble members show diverse patterns in several winter months. It could be that the modelled signal is smaller than in the real world, as suggested by the reviewer, and/or that the modelled variability is too large, both of which would affect the signal-to-noise ratio and hence the detectability of the solar cycle response in the UM-UKCA ensemble. However, UM-UKCA captures the major components of stratospheric variability, including sudden stratospheric warmings and an internally-generated QBO, which makes the latter case less likely. It could be that the reanalysis record is too short to confidently diagnose the solar cycle signal in the high latitude stratospheric zonal winds, and that the reanalysis results in Section 6 are important for interpreting the solar cycle signal found in reanalysis datasets (which by definition represent only a single realisation of the real atmosphere currently for a few decades).

Nevertheless, we have mentioned in Section 4 the somewhat earlier timing of the NH dynamical response in UM-UKCA compared with the reanalysis best estimate and suggested this could potentially be related to the biases in the modelled climatological mean state.

**(8) Most of the manuscript beginning with section 4.2 could be criticized as being either over-analysis of a deficient model or an investigation of issues that are mainly of academic interest. Can the authors find ways to delete or at least shorten some of this material?**

As suggested by the reviewer, we have shortened Section 4.2.

Regarding Sections 5-6, we believe these sections provide a valuable insight into the role of analysis method and interannual variability for the detectability of the solar cycle within a 'perfect model' framework. Of course the model is not perfect and have some shortcomings as discussed above, but we still believe the results provide valuable information, e.g. about length of datasets required, ensemble sizes, that will be helpful for both future observational and model analysis studies. These issues have not yet been properly discussed in the literature, but are particularly relevant given the relatively short length of the satellite record compared with the quasi 11-year period of the solar cycle. The results also complement other important community efforts such as the SPARC SOLARIS-HEPPA working group on Methodological Analysis.

References:

See the references list in the manuscript, as well as:

Long, C. S., Fujiwara, M., Davis, S., Mitchell, D. M., and Wright, C. J.: Climatology and interannual variability of dynamical variables in multiple reanalyses evaluated by the SPARC Reanalysis Intercomparison Project (S-RIP), Atmos. Chem. Phys., 17, 14593-14629, https://doi.org/105194/acp-17-14593-2017, 2017.

Response to Referee #2

**This is an interesting and very well written study on solar 11-year signatures in different atmospheric parameters (in the troposphere, stratosphere and lower thermosphere) based on model simulations with the UM-UKCA model. The paper does not present any spectacular new results on atmospheric effects of solar variability at the 11-year scale, but it is an interesting contribution to the field and should eventually be published in my opinion. An important aspect of the study is the fact that two different analysis techniques (i.e., a composite analysis and multi-linear regression) are applied and the differences in the results are studied and discussed.**

We thank the reviewer for their positive review and the constructive and helpful comments for improving the manuscript. We reply to their comments below in blue.

**I ask the authors to consider the following comments:**

**Page 2, line 15: "The spectral distribution of solar irradiance is commonly referred to as the spectral solar irradiance (SSI)." I disagree, this is not the correct meaning of solar spectral irradiance. SSI has the units W / m2 / nm, i.e. spectral irradiance. It is the power of electromagnetic radiation per unit area and per spectral interval. SSI at a certain wavelength can also be determined or calculated without considering the spectral distribution of the entire spectrum.**

This sentence has been removed and replaced by: "The variation in solar spectral irradiance (SSI) as a function of wavelength is important for determining the atmospheric response to the solar cycle."

**Page 5, line 29: "In the Fast-JX photolysis scheme, the change in partitioning of solar irradiance . . ." I find the phase "change in partitioning" a little misleading, because it's not only the partitioning that's changing. The overall TSI changes as well.**

We have carefully chosen the phrase "change in partitioning of solar irradiance into wavelength bins" in light of the fact that the Fast-JX bins cover wavelengths from 177-850 nm and, thus, do not encompass the full solar spectrum. We therefore defer from referring to TSI changes in Fast-JX as this excludes a part of the full spectrum.

**Page 5, last line: "At pressures less than 0.2 hPa, i.e. where photolysis rates are calculated using the look up tables, the 11-year solar cycle variability is reflected in the TSI change only, with no modulation of SSI." I don't really understand this statement. If TSI is changed, then SSI (in a given spectral interval) must change as well. You probably mean that the spectral distribution of the solar irradiance spectrum is not changed, right? See also my comment on the meaning of SSI above.**

The reviewer is correct. We have clarified the text accordingly.

**Page 6, line 22: ".. but a sparse horizontal sampling (Soukharev and Hood, 2006; Hood et al., 2015; Tummon et al., 2015)." I think it's more appropriate here to cite one of the original instrument or algorithm papers, rather than papers that "only" use the data. Sparse geographical coverage was always known to be a disadvantage of solar occultation observations.**

We have now added references to Damedeo et al. (2013, 2014).

**Page 7, equation 5 (and the equation in the supplement): The choice of the offset and trend terms does not make sense to me. The offset is just a number, right? Why does it have to be represented by a product of two numbers. This is not necessary and only makes things more complicated. I doubt that the function is implemented in this way in your fitting routine – this would not lead to stable results. Also, the trend term trend(t) is simply "t", right? If yes, then it should be written that way.**

The MLR code does indeed represent the offset and trend terms as products of two numbers, and this approach has been previously used in the literature (e.g. Chapter 8 in SPARC CCMVal, 2010; Kunze et al., 2016).

As discussed in Sect. 2.4 the calculation of the yearly mean response by our MLR model is carried out using monthly-mean input data, and the seasonal cycle is accounted for by expanding the regression coefficients (i.e. the 'b' terms) into pairs of sine and cosine functions.

Thus, the term "b(offset)·offset" accounts for the 12-month climatology (i.e. 12 different values depending on a month). Similarly, the "b(trend)·trend(t)" is the trend term modified the annual cycle.

The trend(t) term is indeed just "t" but we would rather stick to the former name for consistency with the other terms in the equation.

**Page 7, line 16: "(here applied 5 times)" Is there a specific reason, why this filter was applied 5 times?**

We have decided to apply the filter 5 times in order to smooth the ENSO timeseries. This choice was made arbitrarily and we believe it would not bear large impact on the diagnosed solar response.

**Page 8, line 14: "However, unlike the yearly mean TSI timseries that forces the model, the timeseries chosen here is that originally recommended for the CMIP5 models" How does this choice affect the results? Ideally, the same solar proxy time series should be used. Please add a brief (qualitative) comment on the expected impact (probably very small).**

This choice is made for consistency since the model is forced with annual mean TSI while the monthly mean observation/reanalysis data will be affected by the monthly variations in solar irradiance. However, given that the amplitude of the 11 year solar cycle is larger than typical month-to-month fluctuations, this signal dominates the analysis and hence the choice of slightly different proxies is unlikely to affect the results.

**Same line: "timseries" -> "timeseries"**

Thank you for spotting this typo. Corrected.

**Section 3: It would be good to show a sample result of the MLR analysis (fit and residual). I have no reason to doubt that the method works well, but it's always good to see a fit example.**

We have added an example of a fit and a residual to the Supplement (Fig. S2), and we refer to it the manuscript.

**Page 10, line 17: "According to the postulated . . ." I think this sentence is incomplete.**

This has been reworded to: "According to the mechanism postulated by Kodera and Kuroda (2002),…"

**Page 11, section 3.2.3: This section focuses more on the (few) similarities between ERAI and the model simulations. However, looking at Figs. 3 and 4 the obvious aspects are the significant differences for both T and the zonal wind response. They should be mentioned/discussed as well.**

As given by the heading to Sect. 3.2.3., this section is about the response in the mid-latitude troposphere. We discuss the broad similarity in the NH, and we also note the differences found in the SH. We do not think there are many other aspects regarding the solar response in the mid-latitude troposphere that needs to be discussed in that section.

**Page 12, line 14: I suggest replacing "The lower altitude of the ozone response" by "The lower altitude of the maximum ozone response"**

Corrected as suggested.

**Page 17, line 10: "we find that the total column ozone responses derived in various regions are somewhat higher for MLR than for composites," Any ideas on the causes of this behaviour?**

We have not investigated this feature in detail. However, it is plausible that the composites are more strongly affected by random dynamical variability, which is particularly important for the determining the total ozone response to solar forcing (Hood, 1997), whereas the MLR explicitly treats the noise as a separate term.

**Page 17, line 28: "observational records such as ERAI" Can one really call ERAI an observational record? It's certainly different from the "pure" observational records such as the SAGE II O3 data set.**

We agree and have changed this to "observational/reanalysis records"

**Page 21, line 18: "Some differences (although not statistically significant) are found in the troposphere and in the tropical lower stratosphere." Did the paper really show that the differences are not statistically significant? Some signatures are statistically significant in one analysis, but not in the other. What does this imply in terms of the statistical significance of the differences?**

We agree that we have not explicitly discussed the issue of statistical significance of the composite-MLR differences. However, in Fig. 2b,c (ozone and temperature), the confidence intervals associated with the individual MLR and composite responses overlap throughout the tropical troposphere and stratosphere, thereby illustrating the differences between the MLR and composite responses in this region are not statistically different. We have further tested the differences between the composite and MLR zonal wind and temperature responses in Fig. 3a-b and 4a-b by looking for regions where the confidence intervals (i.e. ±2 standard errors) around the individual composite and MLR responses do not overlap (and, hence, where the composite-MLR differences are statistically significant), not shown. The results show that the differences are indeed mostly not statistically significant.

Given the reviewer's concerns, we have changed the sentence in question to 'Some apparent differences (although mostly not highly statistically significant)…'

**Figure 2, caption and title of panel a): "heating rates response" -> "heating rate response"**

Corrected.

**Page 43, Table 1, lines 3 and 4: Both lines list the same spectral interval (320 – 690 nm). Is this intended? If yes, the exact meaning of these two lines (and their difference) is not clear to me.**

This is intended and relates to the design of the model shortwave radiation scheme. It includes one band to account for absorption by ozone and a separate band to treat the overlapping absorption between ozone and water. More details can be found in Zhong et al. (2008), Cusack et al. (1999) and Edwards and Slingo (1996).

**Simulating the atmospheric response to the 11-year solar cycle forcing with the UM-UKCA model: the role of detection method and natural variability**

[revised manuscript text omitted]

The variation in solar spectral irradiance (SSI) as a function of wavelength is important for determining the atmospheric response to the 11-year solar cycle. Given a typical change in total solar irradiance (TSI) over the 11-year solar cycle of ~1 Wm$^{-2}$, the associated percentage irradiance variability in the visible and infra-red parts of the spectrum is relatively small; the variability in the ultra-violet (UV) region, however, is larger (Fig. S1, Supplement). The spectral distribution of solar irradiance is commonly referred to as the spectral solar irradiance (SSI). 
[revised manuscript text omitted]
 methodss, respectively. The short wave heating rates increase due to the enhanced absorption of solar radiation by ozone, caused by the combination of the increased UV flux and elevated ozone levels (Haigh, 1994). Notably, this solar-induced modulation constitutes a relatively small fraction of the absolute short wave heating rates in this region (~1.5 % near 50 km). The magnitude
15   of the MLR-derived maximum is somewhat larger than in the composites, although the two are not significantly distinguishable, in a statistical sense, given the estimated uncertainties in each.

**3.2 Temperature and zonal winds**

**3.2.1 The tropical upper stratospheric temperature response**

The corresponding yearly mean tropical temperature response to the solar cycle forcing is shown in Fig. 2b, compared with
20   the ERAI reanalysis. The MLR tropical temperature response in UM-UKCA maximises above the stratopause at ~0.8 K/$Wm^{-2}$. In comparison, the tropical mean ERAI response maximises in the upper stratosphere at ~1-1.1 K/$Wm^{-2}$ (in agreement with previous ERAI studies, e.g.: Mitchell et al., 2015b; Hood et al., 2015; Kodera et al., 2016). Compared to ERAI the UM-UKCA simulated temperature maximum thus occurs at higher altitudes and is ~25% smaller.

A number of factors exist that could potentially explain the apparent underestimation of the maximum tropical temperature
25   responsemaximum in UM-UKCA as compared with ERAI. First, there are large uncertainties in the reanalysies datasets in the upper stratosphere and above due to the scarcity of long-term measurements and, thus, inefficient assimilation and bias correction (McLandress et al., 2014; Long et al., 2017Mitchell et al., 2015a; 2015b). As a result, large differences exist between reanalyses in both the structure and magnitude of the upper stratospheric/lower mesospheric temperature solar response to the solar cycleseen in various reanalysis products (Mitchell et al., 2015b). In fact a smaller temperature response was estimated
30   from the stratospheric sounding unit satellites (up to ~0.6-0.7 K/100 SFU, SPARC, 2010; up to ~1 K/125 SFU, Randel et al.,

2009; recall that 1 Wm$^{-2}$ ≈ 100 SFU, Section 2.4), although this could be related to their relatively poor vertical resolution (Gray et al., 2009).

Another possible factor contributing to the smaller temperature response in UM-UKCA compared to ERAI is the broadband shortwave heating scheme, with only one band in the UV and two in the visible parts of the spectrum (Section 2.1.2). Nissen et al. (2007) showed that decreasing the number of spectral bands in the UV/visible range from 49 to 6 can result in an underestimation of the stratopause shortwave heating response to the 11-year cycle by ~20 %. In addition, only the absorption of solar radiation by ozone is considered in the first (UV) shortwave spectral band (Table 1), thereby neglecting the absorption by molecular oxygen. Also, limiting the shortwave heating scheme to the wavelengths higher than 200 nm excludes changes in the mesospheric absorption by oxygen near the Lyman-α line (121.6 nm), where percentage irradiance changes during the solar cycle can be particularly large (Lean, 2000; Nissen et al., 2007).

Furthermore, the model has used a relatively modest modulation of SSI. There has been considerable uncertainty associated with the solar cycle modulation of SSI due to the shortage of long-term satellite measurements, with marked differences between the individual available datasets (e.g. Harder et al., 2009; Dhomse et al., 2013; Ermolli et al., 2013). We also note that whilst being consistent with the design of the HadGEM2-ES model (Jones, et al., 2011), the current implementation of the solar cycle forcing in the model's radiation scheme results in an underestimation of the UV changes in the 200-320 nm band by around ~20 % as compared to the more recent CMIP5 SSI recommendations (Section 2.1.2).

Nonetheless, we stress that even with the use of identical imposed SSI relatively broad ranges of modelled solar cycle temperature responses have been reported (e.g. Austin et al., 2008; SPARC CCMVal, 2010; Mitchell et al., 2015a). SPARC CCMVal (2010) also showed that model performance in simulating the direct radiative response to a change in solar irradiance alone could not fully account for the spread of simulated temperature responses, suggesting some contribution of indirect dynamical processes. This aspect will be investigated further in Sect. 5, where the solar response found from the individual ensemble members is considered. Lastly, the differences in stratospheric temperature responses could be related to UM-UKCA solar ozone responses, which and its relation to the tropical temperature response, is is discussed in the next sSection 3.3.

[revised manuscript text omitted]
 (2002); h this is consistent with the small warming simulated in this region. However, the UM-UKCA tropical $\overline{w}^*$ response, particularly at the altitudes of ~100-30 hPa, is very small and not highly statistically

significant, indicating that the postulated mechanism is not robustly reproduced in the model. A more statistically robust response was found in the mid-latitudes, possibly indicating a relative shift in the downwelling region towards the extratropics. In December and January, the sign of the model responses reverses: the enhanced wave breaking under higher solar cycle forcing strengthens polar stratospheric downwelling (Fig. 9). Regarding the tropical response, although its sign is consistent with the strengthened BDC, the response is again (as in autumn) very small and not significant.

[revised manuscript text omitted]

---

## Author Response (AR2)

Thanks to the authors for their replies and for making some minor changes to the
previous manuscript version. However, the manuscript still does not convey
sufficiently the very clear qualitative and quantitative differences between the
model's stratospheric response to 11-year solar forcing and that which has been
derived from the available observational data.

We thank the reviewer for evaluating our revised manuscript. We hope our responses below
address their outstanding concerns, particularly around the comparison between the modelled 11-
year solar signals and the observations/reanalysis. Our replies to the points raised are given below
in blue.

It is not noted anywhere that
at least a few of the CMIP-5 models (some of those with coupled oceans as well
as interactive ozone chemistry) do a much better job of simulating the observationally
estimated response, including the zonal wind response in the upper stratosphere
that is found in both the northern and southern hemispheres.

To address this we have added a sentence to the introduction that reads: "While some CCMs
simulate solar responses that broadly resemble the solar cycle signals derived from
observations/reanalyses, some studies have found a marked spread of the solar responses between
different models (see e.g.: Austin et al., 2008; SPARC, 2010; Hood et al., 2015; Mitchell et al.,
2015a; Maycock et al., 2018)."

The results in sections
5 and 6 regarding the role of analysis method and interannual variability on
the detectability of the solar cycle signal in observations are not really
new although I can agree with the authors that more discussion of these issues
is needed in the literature. The results in section 6 do not take into consideration
the lack of realism of the model and the possibility that interannual variability
in the model may be greater than that in the actual atmosphere.

We are pleased the reviewer agrees that the topic of analysis method and interannual variability
for solar signal detection is an important issue that warrants greater discussion in the literature.
This was the primary motivation for Sections 5 and 6 our manuscript. We address the specific
points raised about the realism of the modelled response in our replies to the comments below.

In view of the
minor changes made to the revision, I went back and forth between recommending
rejection and major revisions and finally settled on major revisions.

(1) The abstract and summary (section 7) still make little mention of the strong
disagreements between the model-simulated responses of stratospheric ozone,
temperature, and zonal wind to 11-year solar forcing and that derived
from observations (here, ERA-Interim and SAGE II). The summary section is virtually
unchanged. The only admission of model deficiencies added to the manuscript
is (abstract): ``... there are some differences in magnitude, spatial
structure and timing of the signals in ozone, temperature and zonal winds.'' There is no
acknowledgment of the very clear qualitative differences between the model response
and that derived from observations in the abstract or summary sections.

We appreciate that the referee asks for even more critical assessment of our results; however, we
would like to first point out the discussion of these differences already found in our manuscript
and further add to that. We have extracted below paragraphs 2-4 of the Summary section
(Section 7) of our revised manuscript. We highlight in yellow the sentences which explicitly

state/discuss the differences between the UM-UKCA model results and observations/reanalysis and add in red further changes implemented in the revised manuscript:

"Regarding the ensemble mean UM-UKCA response, the enhanced solar cycle activity increases the stratospheric shortwave heating rates and temperatures. The resulting yearly mean warming maximises near the tropical stratopause at ~0.8 K/Wm$^{-2}$. The response occurs at higher altitudes and is ~25% smaller than that derived from ERAI. A number of  factors possibly contributing to this  underestimation of model temperature response was  identified (Section 3.2.1): i) the relatively broadband shortwave heating scheme; ii) the lack of $O_2$ absorption in the radiation scheme; iii) the underestimation (~20%) of UV (200-320 nm) variability compared with CMIP5; iv) the use of a modest SSI variability (Ermolli et al., 2013); v) uncertainties in the reanalysis (Mitchell et al., 2015b). For ozone, the UM-UKCA model simulates a yearly mean tropical ozone increase of up to ~2.0-2.5 %/Wm$^{-2}$ in the mid- stratosphere. . Unlike the more peaked and locally stronger SAGEII ozone response, the maximum model response is weaker, more horizontally uniform and occurs at lower altitudes. We note that  differences exist between the  temperature and ozone responses derived from various observational and reanalysis datasets  (Soukharev and Hood, 2006; Dhomse et al., 2013; 2016; Mitchell et al., 2015b; Maycock et al., 2016; 2018). Averaged over the globe, the yearly mean total ozone column response simulated in UM-UKCA was estimated (MLR) to be of ~6 DU/Wm$^{-2}$.

The analysis did not find a yearly mean secondary temperature or ozone maximum in the model in the tropical lower stratosphere as seen in the reanalysis. This may be related to differences in the associated dynamical responses in both hemispheres, as manifested by the absence in the model of the yearly mean strengthening of the extratropical stratospheric jets seen in ERAI. Despite that, we do find a small warming of up to ~0.1-0.2 K/Wm$^{-2}$ in the NH mid-latitude troposphere alongside the associated weakening of the NH subtropical jet on its equatorial side. This tropospheric/lower stratospheric response is in broad qualitative agreement with the reanalysis and suggests a contribution of the solar signal in the prescribed SSTs/sea-ice, as found by Misios and Schmidt (2013).

In accord with the mechanism postulated by Kuroda and Kodera (2002) and Kodera and Kuroda (2002), the enhancement of the horizontal temperature gradient under increased solar cycle activity strengthens and cools the NH stratospheric vortex in autumn. The simulated response extends to the troposphere in November. A sign reversal of the modelled stratospheric response occurs in mid-winter (January). The modulation of the NH polar jet in the model is associated with consistent changes in planetary wave propagation and, at least in the high latitudes, the meridional overturning circulation. In general, the evolution of the NH dynamical solar response in UM-UKCA during autumn and winter shows some broad resemblance to that seen in ERAI. However, the model shows earlier  timing of the  responses , which could be related to the positive bias in the model's zonal wind climatology and/or too weak SSI forcing. In addition to the different timing, the simulated westerly response diagnosed from monthly mean data appears first at higher latitudes than in ERAI, thereby not clearly reproducing the poleward/downward propagation. In general ,

any (monthly-mean) westerly anomalies near the subtropical stratopause in UM-UKCA are much weaker and shorter-lived than in ERAI; the UM-UKCA model ensemble also does not reproduce the westerly anomaly observed in the NH mid-/high latitudes in mid-/late spring (April onwards)."

\*\*\*

Regarding the abstract, we recognise the short and concise nature of abstracts in general, and thus we believe that referring to each individual feature is beyond its scope. However, we have reworded the abstract as follows: "The UM-UKCA model produces a statistically significant response to the 11-year solar cycle in stratospheric temperatures, ozone and zonal winds. However, there are also differences in magnitude, spatial structure and timing of the signals compared to observational and reanalysis estimates. This could be due to deficiencies in the model performance, and so we include a critical discussion of the model limitations, and/or uncertainties in the current observational estimates of the solar cycle signals." This rewording and change in emphasis addresses the reviewer's concerns about making clear the differences between the modelled and observationally estimated signals

In the reply to this criticism, it is argued that, in Fig. 2, "the uncertainties in the best estimates of temperature and ozone responses in the tropics are overlapping throughout most of the stratosphere." This is not a convincing answer for three reasons. First, there is a qualitative difference in the altitude structure of the ozone and temperature responses estimated from observations and that estimated in the model. The observations (both ERA-Interim and SAGE II) indicate maximum responses in both the tropical upper and lower stratosphere with a minimum near 30-35 km. In contrast, the model tropical temperature response declines monotonically with decreasing altitude while the model ozone response has only a broad maximum centred near 35-38 km. The model has no lower stratospheric response and the response in the upper stratosphere is significantly weaker than that derived from SAGE II data.

We agree that there are qualitative differences between the model and observations/reanalysis as the reviewer points out. The sections of highlighted text above show that we do state all of these above-mentioned differences by the reviewer in our Summary section (Section 7), and we do discuss them in detail in the relevant results section (Sections 3.2.1; 3.2.2; 3.3.2). However, given the reviewer's concerns, we have further improved on Section 7 (as described above) and the relevant results sections (Sections 3.2.1; 3.2.2; 3.3.2). The fact that there are overall differences between the model and observations/reanalysis is also clearly stated in the revised abstract.

Second, the fact that the model error bars overlap with the observational error bars is not a reason to accept the model results because it is the best estimate by the model (the mean of that estimated from the ensemble members) that should be compared to the observational error bars. If an infinite number of ensemble members was available, the error of the mean would decrease to zero.

We do not fully agree with this point.. Firstly, the UM-UKCA ensemble mean MLR/composite results are produced by combining the data from the 3 individual 1966-2010 ensemble members into one 135-year long record prior to the analysis, and the MLR/composite analysis is performed on the combined dataset (Section 2.4). For MLR, it is not the same as averaging the results from the MLR analysis applied separately to the three individual ensemble members (in fact this would be wrong to do so). While the resulting ensemble mean values could be similar, the resulting

standard errors (obtained as the square roots of the diagonal elements of the covariance matrix) would not converge on zero in the presence of infinitely long timeseries. They would converge to values representing an internal variability of the stratosphere. (We do note that the standard errors of our composite responses would decrease to zero for an infinitely long timeseries.)

Secondly, we disagree that it is only the mean of the model results that should be compared to the observational error bars. This would be true if the model was run in a specified-dynamics configuration (i.e. where the model meteorology was constrained towards the observations) and the observational error bars accounted for any instrumental or assimilation uncertainties. However, this is not the case here as the model is run in a free-running configuration and, as such, the model error bars account for the variability of the diagnosed solar signal due to simulated internal variability of the stratosphere. The observational/reanalysis error bars shown in the manuscript, on the other, do not take into account instrumental/assimilation uncertainties at all, but as with the model reflect the effects of the stratospheric internal variability on the signal detection. We believe that the internal variability in the stratosphere (which can be particularly large in certain regions, e.g. in the high latitudes) needs to be accounted for when considering both the model and observational/reanalysis results, and thus it is important and fully justified to consider both the model and observational/reanalysis error bars.

The mean profile based on 3 ensemble members falls
outside of the observational error bars at 2 hPa for temperature and near 30 km for ozone.

Indeed, but we do already state both the underestimation of the upper stratospheric temperature response compared to ERAI and the fact that the modelled ozone response has a different, more horizontally-uniform structure to SAGE II in the Summary Section 7 (see paragraph 2, lines 19, 24-25) of our revised manuscript (see also Sect. 3.2.1, paragraph 1; Section 3.3.2, paragraph 1). We would like to nonetheless point out here that we do not propagate errors here and our observational/reanalysis error bars are estimated from the final data product only (and as such do not take into account any instrumentational/assimilation uncertainties).

Third, Figure 2 is a very smoothed (latitudinally averaged) comparison that makes the model
results look better than they really are. The actual latitudinal structure is shown in
Figures 3-5 where it is seen that there is a clear qualitative difference between the
tropical stratospheric solar response of ozone or temperature simulated in the model
and that estimated observationally.

We agree that there are qualitative differences between the model results and observations/reanalysis. These are discussed thoroughly in the results Sections 3 (and 4 for zonal wind), clearly listed in the Summary Section 7, and the overall fact that there are differences between the model and observations/reanalysis pointed out in the abstract.

In the reply, it is argued that observational
uncertainties nevertheless allow use of the model results to ``provide a valuable insight into
the role of detection method and interannual variability for the detected
solar cycle signal/response that has not been widely acknowledged in the previous
literature.'' However, the presented model results do not give any significant
new insights beyond what is already well known to data analysts and modelers
(see comments 5 and 6 below).

Please see our responses to comments 5 and 6.

(2) The weight of the evidence still indicates an underestimation by the

model of the upper stratospheric ozone and temperature responses. Accepting
the argument that it is the MLR-derived model ozone response of 1.5% at 45 km in
Figure 5b (rather than 1.0%) that is relevant, the ozone response derived from SAGE II
data within 10 degrees of the equator is still more than 3 per cent and extends up to 50
km, whereas the model mean value is between 1 and 1.5 per cent between 45 and 50 km.
The smoothed comparison shown in Figure 2c is unconvincing for reasons given in comment
(1). The ERA-estimated tropical temperature response in Figure 3c exceeds 1.1 K at
altitudes as low as 43 km while the model temperature response in Figure 3b maximises
at 0.9 K near 53 km and is less than 0.6 K at 43 km. Accepting the reply that the
6 spectral bands apply only to the model's shortwave radiative transfer scheme,
there remains a concern that the assumed irradiance variation may be too small.
On p. 5, line 31 of the new manuscript version, it is noted that the assumed irradiance in
the main UV band is 20\% less than that recommended in the CMIP-5 SSI specifications.

We agree that the specification of SSI in the model's radiation code is important for the simulated temperature response, and the implication of this limitation is discussed in Section 3.2.1 of our manuscript. However, this will not be the case for the photolysis scheme (and thus ozone response) as the SSI change in FastJX does follow the fit to the CMIP5-recommended irradiance for the years 1981-1986 (Section 2.1.2). We apologise for the confusion.

Several CMIP-5 models with interactive ozone chemistry have simulated stronger upper
stratospheric ozone responses (about 2% at 45 km) and temperature responses
(about 1 K at 45 km), suggesting that the adopted SSI variation may be too weak.
But even the CMIP-5 recommendations could be too small because, as reviewed by Ermolli
et al. (ACP, v. 13, p. 3945, 2013), there remain significant differences between proxy
solar spectral irradiance models. If the SSI variation is larger than the CMIP-5
recommendations at wavelengths that affect O2 photolysis and radiative heating near
45 km, then the existing model code would produce larger ozone and temperature responses
consistent with those derived observationally. There is no acknowledgment of this fact
in the manuscript.

We did attempt to acknowledge this with respect to the underestimation of the upper stratospheric temperature response. The top of paragraph 3, Section 3.2.1 of our revised manuscript reads: "Another factor that is likely to be important for the magnitude of the upper stratospheric temperature response is the fact that the model has used a relatively modest modulation of SSI. There has been considerable uncertainty associated with the solar cycle modulation of SSI due to the shortage of long-term satellite measurements, with marked differences between the individual available datasets (e.g. Harder et al., 2009; Dhomse et al., 2013; Ermolli et al., 2013)."

The reviewer is correct though that we did not acknowledge this with regard to the ozone response, and so we have added the following sentence to Section 3.3.2: "As noted in Section 3.2.1., there are large differences between the different SSI datasets (e.g. Ermolii et al., 2013) and the UM-UKCA model is forced with relatively modest modulation of SSI at the lower end of the current estimates".

(3) The lack of a tropical lower stratospheric response to 11-year solar forcing
is still a deficiency of the model. Thanks for adding the sentence to the
Introduction noting that coupling between the lower stratosphere and tropical
tropospheric convection could be important for producing or amplifying the tropical
lower stratospheric response (lines 18 and 19 on p. 3 of the revised version). However,
the authors' reply that using prescribed SSTs accounts sufficiently for this coupling
is unconvincing. There is no tropical lower stratospheric response in the model.
Prescribed SSTs will not account for any coupling via the MJO, for example. Several

CMIP-5 models with coupled oceans produce a tropical lower stratospheric response even
when time periods without significant volcanic eruptions are analysed.

We agree with the reviewer that using interactive ocean is important for many reasons, and the last sentence in our Summary Section 7 reads: "The results indicate the need to use coupled atmosphere-ocean models in order to fully capture the impacts of the solar cycle forcing on the tropospheric climate." See also results in Section 3.2.3, as well as the last paragraph in Section 6. While we believe that using prescribed observationally-derived SSTs would include some effect of the solar signal present in SSTs on the troposphere, we recognise that this may not be sufficient. We have now added a sentence to Section 3.2.2: "Possibly, the coupling of the ocean and tropical convection, which may also play a role in influencing the tropical lower stratosphere (e.g. Yoo and Son, 2016), is also not adequately represented in the model set-up with prescribed (although observationally derived) SSTs." We note, however, that most CMIP5 models have a poor representation of the MJO (e.g. Ahn et al., 2017) so even models with coupled oceans are unlikely to properly capture coupling of the solar cycle to the lower stratosphere via the MJO. The reviewer has not included any reference to specific literature, so unfortunately we are unable to verify or reply to their statement about CMIP5 models simulating a lower stratospheric response outside of volcanic periods.

(4) The lack of a zonal wind response at northern midlatitudes in December,
January, and February remains a major shortcoming of the model. Without such
a response, the model is incapable of simulating the solar dynamical signal.
There is no admission of this important fact in the manuscript.
While the manuscript notes the too-early timing of the modeled response in
November, there is still no mention of the latitudinal bias of the response
such that it is found at 60N in the model but at 30-40N
in the observations.

While the reply emphasises uncertainties in the observations,
it is important to note that the strong zonal wind positive response in the midlatitude
upper stratosphere and lower mesosphere that is found in DJF is also found with even
larger amplitude at southern midlatitudes (30-40S) in JJA (see, e.g.,
Figure 4 of Crooks and Gray, J. Climate, v. 18, p. 996, 2005). The existence of a
corresponding zonal wind
response during austral winter provides empirical evidence that the response is real
and must be simulated realistically in a model if it is to be used to evaluate
the detectability of the solar cycle signal.

We now acknowledge the latitudinal differences in the NH zonal wind response compared to ERAI. In Section 4.1.3, we have added the following lines: "In addition to the different timing of the zonal wind response to ERAI, the simulated westerly response diagnosed from monthly mean data appears first at higher latitudes than in ERAI, thereby not clearly reproducing the poleward/downward propagation seen in the reanalysis (although the model does simulates a significant dynamical response to the imposed solar forcing at high latitudes comparable in magnitude to that diagnosed in ERAI)". In Section 7, we have added: "In addition to the different timing, the simulated westerly response diagnosed from monthly-mean data appears first at higher latitudes than in ERAI, thereby not clearly reproducing the poleward/downward propagation."

We note that the fact the model simulates a significant dynamical response to the imposed solar forcing at high latitudes comparable in magnitude to that diagnosed in ERAI means we can still use the model to investigate the detectability of the solar signal even if particular details of it may differ from that estimated from the reanalysis.

The comparison shown in Figure R1 of the
reply does not alleviate this concern because the comparison is done at 60N
where the observationally estimated response is weak or non-existent. The error bars
on the modeled zonal wind response are also quite large because the ensemble includes
only 3 members. Again, the requirement should be that the model's best estimate
(the mean of the three realisations) should fall within the observational error bars.
But the comparison should be made at the latitude where the observationally estimated solar
signal is found.

We first clarify, as in the response to point 1 above, that the model error bars reflect the uncertainty on the MLR applied to the combined ensemble of one 135-year long record (and not the variance between three mean values derived from an average of separate MLR applied to each ensemble member, as the reviewer suggests). The error bars for both the model and reanalysis/observations therefore reflect the uncertainty in the signal detection for the two timeseries of different length and, hence, they should both be considered when comparing the model and reanalysis/satellite data results. The 135-year timeseries from the UM-UKCA ensemble is significantly longer than the ~30 years available from the ERAI reanalysis and as such one expects the error bars to be smaller for the model.

We have redone Figure R1 for 30N, 35N, 40N and 45N (see below). As it was for 60N, there is still a large degree of overlap between the error bars around the winter UM-UKCA and ERAI zonal wind responses.

[Figure]

[Figure]

Fig. R1. November to April zonal mean zonal wind response [ms$^{-1}$/Wm$^{-2}$] at selected latitudes diagnosed using MLR for UM-UKCA ensemble (blue) and ERA-I (red). The error bars denote ±2 standard errors of the mean response.

(5) The results in section 5 regarding the role of analysis method (compositing versus MLR) are well known and do not provide any new insights regarding the detectability of a given forcing signal. It is well known that MLR results for any quasi-periodic signal (ENSO, QBO, or solar cycle) will differ from compositing results because the MLR method accounts approximately for other sources of interannual variability. For this reason, nearly all observational analyses use MLR. It is also well known that the differences will be larger in regions where interannual variability is larger (i.e., in the tropical troposphere and at high latitudes).

We believe that our results present interesting differences in the comparison of MLR and composite analysis by region which have not been described before. For example, we show that the solar cycle signals derived from MLR and composites are in fact in good agreement with each other in the high latitudes contrary to what might be expected owing to the large interannual variability there. Secondly, we have shown qualitative difference between the MLR and composite temperature responses in the tropical troposphere. This is an interesting and relevant finding, in particular in the context of potential coupling between the solar cycle forcing and other atmospheric forcings and modes of variability (e.g. QBO, ENSO) that has been discussed in the literature (see e.g. Salby and Callaghan, 2000; Gray et al., 2004; Pascoe et al., 2005; Labitzke et al., 2006; Haigh and Roscoe, 2006; Camp and Tung, 2007; Kuroda, 2007; Lu et al., 2009; Calvo and Marsh; 2011, Roy and Haigh, 2011). We are unaware of previous literature that makes these specific points when comparing MLR and composite results, but if the reviewer does know of such studies we are happy to include additional references in our manuscript.

(6) The results in section 6 (comparison of yearly mean temperature, ozone, and zonal wind results derived by MLR for the three ensemble members (Figures 10-12)) are of interest but do not take into consideration the possibility that interannual variability in the model may be greater than that in the atmosphere. There is no mention of this possibility in the manuscript.

We have added a paragraph to Section 6 that says: "Possibly, the solar cycle signal in the UM-UKCA model is smaller than in the real world, owing to e.g. the relatively weak SSI forcing (Sections 3 and 4). The model integrations also show higher interannual variability (i.e. standard deviation) of the NH zonal wind in parts of the stratosphere, e.g. the upper stratosphere around 60N in autumn/winter, than ERAI (not shown). Both of these effects will impact on the signal-to-noise ratio and the detectability of the solar cycle signal (Scaife and Smith, 2018)." We also put more emphasis in the text that the results/conclusions of Section 6 apply to the model(s).

More generally, the issue whether climate models realistically capture internal variability has implications for every single detection study using models in climate science and is beyond the scope of this work (see e.g. Scaife and Smith ,2018, for a recent review).

It is well known that a
large number of ensemble members is needed to extract with high confidence a given
geophysical signal from model data. The number of members and total record length
considered here (three
45-year simulations representing a total of 135 years subjected to MLR analysis)
is too small.

As noted in the reply to the previous comment, the validity of this statement will depend on the amplitude of the signal being extracted and how orthogonal a forcing is to other known drivers. The statement that 135-years of simulation is not sufficient to extract the solar signal would appear to potentially contradict the reviewer's confidence in extracting with high confidence the solar signal from much shorter (30-40 years) observational based records. The length of simulations used here is considerably longer than many published model studies evaluating solar-climate interactions (e.g. Haigh, 1999; Kuroda et al., 2007; Matthes et al., 2004; 2010).

The sentence added to section 6 (lines 14-17 on p. 19 of version 3)
(``From a modelling perspective, it is therefore crucial that the impact of the
solar cycle forcing on climate is studied with sufficiently long simulations and
that the current observations and reanalysis records for the stratosphere are
interpreted carefully, bearing in mind the relatively small number of degrees of
freedom they represent." is certainly true but is there anything new here?

We believe this point is important to stress here in the discussion as it is something not sufficiently discussed in the literature in the context of the solar cycle impacts. (Note we have now removed the second part of this sentence in line with the reply above to make the conclusions of Section 6 more model-oriented.) As noted above, we present new findings on the comparison of the MLR and composite methods as well as new results from the UM-UKCA model compared to reanalysis and observations.

(7) There is no discussion in the manuscript about the need to validate the model
using observational estimates of the atmospheric response to 27-day solar forcing
and the stratospheric QBO before applying it to evaluate the effects of 11-year
solar forcing. The observational uncertainties for these shorter-term forcings
are much smaller than for the 11-year solar forcing problem. Although 27-day
solar UV variability has been weaker during the last few solar cycles, there is
ample satellite and reanalysis data available for the 1979-1993 period when
27-day variability was stronger and atmospheric effects were easier to measure.
It would be straightforward to add daily resolution of the SSI forcing to the model
and investigate its ability to simulate observed atmospheric responses
on this time scale. Probably only 3 ensemble members would be sufficient to
accurately determine the model responses with very small error bars. Similarly, if
the model simulates a QBO, it should be straightforward to determine whether there
are any tropospheric consequences and whether these agree with observational
constraints (e.g., Yoo and Son, 2016).

The model is forced with yearly-mean TSI and as such does not include the 27-day solar forcing. Inclusion and evaluation of this forcing in the model is beyond the scope of this manuscript; however, we agree with the reviewer it would be an interesting avenue to pursue in the future.

[revised manuscript text omitted]